# PRDM9 forms a trimer by interactions within the zinc finger array

Theresa Schwarz[1], Yasmin Striedner[1], Andreas Horner[1], Karin Haase[1], Jasmin Kemptner[2], Nicole Zeppezauer[1], Philipp Hermann[3], Irene Tiemann-Boege[1]

**PRDM9 is a trans-acting factor directing meiotic recombination to specific DNA-binding sites by its zinc finger (ZnF) array. It was suggested that PRDM9 is a multimer; however, we do not know the stoichiometry or the components inducing PRDM9 multimerization. In this work, we used *in vitro* binding studies and characterized with electrophoretic mobility shift assays, mass spectrometry, and fluorescence correlation spectroscopy the stoichiometry of the PRDM9 multimer of two different murine PRDM9 alleles carrying different tags and domains produced with different expression systems. Based on the migration distance of the PRDM9–DNA complex, we show that PRDM9 forms a trimer. Moreover, this stoichiometry is adapted already by the free, soluble protein with little exchange between protein monomers. The variable ZnF array of PRDM9 is sufficient for multimerization, and at least five ZnFs form already a functional trimer. Finally, we also show that only one ZnF array within the PRDM9 oligomer binds to the DNA, whereas the remaining two ZnF arrays likely maintain the trimer by ZnF–ZnF interactions.**

## Introduction

In most mammals, including humans and mice, the meiosis-specific protein PR-domain containing protein 9 (PRDM9) was identified to play a key role in regulating and determining the location of recombination hotspots (Baudat et al, 2010; Myers et al, 2010; Parvanov et al, 2010; Brick et al, 2012; Pratto et al, 2014). PRDM9 is a multi-domain protein expressed in prophase I in ovaries and testis (Hayashi et al, 2005; Hayashi & Matsui, 2006) that recognizes DNA target motifs and directs double strand breaks (DSBs) to these target sites. Four functional domains have been described for the PRDM9 protein: the Kruppel-associated box domain (KRAB), an SSX repression domain (SSXRD), a subclass of the SET (PR/SET) domain, and the C-terminal zinc finger (ZnF) array. All four domains of PRDM9 play an important role in the placement of DSBs at hotspot targets recognized by the ZnF array. Over

evolutionary time, species have either lost the complete full-length *Prdm9* gene, are missing one of the four domains, or have non-functional changes. In those species lacking a functional PRDM9, DSBs occur at PRDM9-independent sites such as transcription start sites or CpG islands, as observed in birds and dogs (Axelsson et al, 2012; Auton et al, 2013; Singhal et al, 2015; Baker et al, 2017; Clement & de Massy, 2017).

The recognition by the ZnF array of a specific nucleotide sequence is the main factor shaping the recombination landscape (Baudat et al, 2010; Grey et al, 2011; Billings et al, 2013; Baker et al, 2014; Walker et al, 2015; Patel et al, 2016; Striedner et al, 2017) with specific target DNA sequences commonly found at hotspot centres in humans and mice (Myers et al, 2010; Brick et al, 2012). This ZnF–DNA interaction is very stable and lasts for many hours, which is important for other PRDM9 domains to carry out their activity throughout the different stages of meiotic prophase I and direct the placement of DSBs in leptotene (Striedner et al, 2017). ZnF arrays vary in the arrangement and number of ZnFs resulting in the activation of different sets of hotspots with an astonishing diversity of ZnF arrays already within species, varying mainly in the amino acids contacting the DNA (reviewed in Tiemann-Boege et al (2017), Paigen and Petkov (2018)). The PR/SET domain methylates H3K4 histones of surrounding nucleosomes resulting in H3K4me3 and H3K36me3 labels (Hayashi et al, 2005; Wu et al, 2013; Powers et al, 2016; Altemose et al, 2017; Grey et al, 2017). The role of H3K4me3 and H3K36me3 in meiosis is not yet fully understood, but both of these epigenetic marks were shown to co-occur at hotspot regions ((Powers et al, 2016) and reviewed by Paigen and Petkov (2018)) and are functionally important in the interaction with components of the DSB machinery, located on the chromatin axis (Imai et al, 2017; Parvanov et al, 2017). In addition, H3K4me3 is associated with an open chromatin structure at DSB targets hypothesized to be important for proper DNA pairing between homologues and recognition, which would be otherwise hidden within nucleosomes (Tiemann-Boege et al, 2017; Heissl et al, 2019). Finally, the N-terminal KRAB domain (together possibly with the SSXRD domain) binds to other protein complexes, such as EWSR1, CDYL, EHMT2 (Parvanov et al, 2017), and CXXC1 (Imai et al, 2017; Parvanov et al, 2017), involved in tethering the target DNA in the loop with the axis

[1]Institute of Biophysics, Johannes Kepler University, Linz, Austria  [2]Red Cross Blood Transfusion Center Upper Austria, MedCampus II, Johannes Kepler University, Linz, Austria  [3]Institute of Applied Statistics, Johannes Kepler University, Linz, Austria

Correspondence: irene.tiemann@jku.at

where proteins of the DSB machinery are located (Kleckner, 2006). Note that PRDM9 interacts with CXXC1, but it is not an essential link for meiotic recombination progression in mice (Tian et al, 2018).

Recently, it has been observed that PRDM9 can form functional multimeric complexes (Baker et al, 2015b; Altemose et al, 2017). How this multimerization affects the activity of PRDM9 is not known, but it could directly affect hotspot activation. To date, observations of PRDM9 multimerization are based on cell systems co-expressing different alleles of PRDM9 with distinct tags (Baker et al, 2015b; Altemose et al, 2017). However, it is not known how many PRDM9 units form the multimer, which is key information to understand how PRDM9 interacts at a molecular level and also influences PRDM9 dosage. In addition, it is still unknown whether different ZnF arrays within a multimeric complex interact with multiple DNA targets.

In this work, we performed an *in vitro* analysis of the DNA–PRDM9 complex using electrophoretic mobility shift assays (EMSAs) to infer the stoichiometry of the active complex. We show that the molecular weight (MW) of a complex can be inferred from its electrophoretic migration distance under nondenaturing conditions. In combination with mass spectrometry, we estimated that PRDM9 forms a trimer when actively bound to DNA. This trimer was observed for two different PRDM9 alleles, PRDM9$^{Cst}$ and PRDM9$^{Dom2}$. Moreover, the trimer formation is mediated within the variable ZnF array and at least 5 of 11 ZnFs are sufficient to form a stable DNA-binding trimer. In addition, using fluorescence correlation spectroscopy (FCS), we also demonstrated that the murine ZnF array already forms a soluble trimer in its unbound state to the DNA. Finally, our data suggest a model in which only one of the ZnF array is involved in DNA binding; whereas, the other two ZnFs likely are involved in protein–protein interactions.

# Results

### Uncoupled binding of PRDM9 with linked successive target sequences

To better understand the binding behaviour of PRDM9 to its target DNA, we used *in vitro* EMSAs. This technique is based on native gel electrophoresis used to analyse a DNA–protein complex visualized by its slower migration compared with free DNA. We designed DNA fragments with one (single-Hlx1) or two adjacent target sites (tandem-Hlx1) derived from the *Hlx1* hotspot known to specifically bind the PRDM9$^{Cst}$ ZnF array (ZnF$^{Cst}$) of *Mus musculus castaneus* origin (Billings et al, 2013; Striedner et al, 2017). For the design of the single and tandem-Hlx1, we considered previous experiments showing that ZnF$^{Cst}$ bound specifically 34 nucleotides, yet unspecific flanking DNA improved the binding (Striedner et al, 2017). Thus, the DNA sequence contained either one or two adjacent 34-bp specific target sites plus 20–23 bp flanking regions (single-Hlx1 or tandem-Hlx1 with 75 bp or 114 bp, respectively), as shown in Fig S1. We analysed the binding of these two DNA fragments to different protein concentrations of ZnF$^{Cst}$ coupled to a maltose-binding domain (hereafter MBP-ZnF$^{Cst}$) in an EMSA titration experiment.

We observed that the DNA with a single binding site formed a complex (shifted band) at low protein concentrations. The intensity of the shift increased with protein concentration saturating the free DNA and forming a complex (Fig 1A), as observed before (Striedner et al, 2017). For the tandem-Hlx1 carrying two consecutive binding sites, two different states of the complex were detected with increasing protein concentrations: a lower shift and a supershift. The lower shift was observed at low PRDM9 concentrations, and as PRDM9 concentrations were increased, a second supershift became visible (Fig 1B). A supershift is observed regularly in EMSAs when a second protein (e.g., antibody against the protein) is incubated with the complex, resulting in a large change of the overall MW slowing the migration of the complex to a supershift (Holden & Tacon, 2011). The observed dynamics can be summarized as follows: the lower shift increased at low PRDM9 concentrations until half of the sites were filled (Fig 1B and D). With further increase in protein, the intensity of the lower shift diminished and was replaced by an increasing supershift. The overall free DNA decayed at the same rate for both the single and tandem-Hlx1 (Fig 1A–D and Table S1).

A quantitative analysis (Fig 1E and F) of this binding showed that the intensity of the sum of the shifts (lower + supershift) was correlated directly with the affinity of the ZnF. We estimated that the tandem-Hlx1 DNA had a similar affinity to the ZnF as the single-Hlx1 ($K_D$ = 35 nM and 48 nM, respectively). Note that these $K_D$ values were slightly higher than those obtained with the same approach in a previous work (24.5 nM ± 2.6) (Striedner et al, 2017). A possible reason for this deviation in the $K_D$ could be the much shorter incubation times used here (60 min versus 90 h) with an effect in the equilibrium states and ultimately the $K_D$ when loading the EMSA.

The most likely explanation for the supershift is the formation of a second PRDM9 complex (double complex) given the similar $K_D$ between these two shifts and the large difference in migration distance between the lower shift and the supershift. To prove that the supershift indeed represents a double complex on the tandem fragment, we did an additional experiment in which we designed a different tandem DNA fragment (232 bp in size), but with the two binding sites separated by a restriction enzyme site (tandem-Hlx1-BamHI) that can be digested into a 75-bp and 157-bp fragment with only one binding site each (Fig S2). When incubating the tandem-Hlx1-BamHI with PRDM9, we again observed a lower shift and a supershift. However, when digesting this fragment after PRDM9 binding, the supershift was gone and instead we observed two lower shifts with the same migration distance as for complexes formed on fragments with one binding site (75 or 157 bp). This experiment demonstrates that the supershift carries two independent complexes that can be separated by a restriction enzyme digest. These results also suggest that the multiple ZnF arrays within a multimer do not interact simultaneously with several DNA-binding sites. Further experiments exploring the interaction of the multimer with several DNA-binding sites are shown in the section "PRDM9 complex binds only one DNA molecule at a time."

### The MW of the PRDM9–DNA complex can be determined by native gel electrophoresis

Native gel electrophoresis can be used to infer the MW of negatively charged, linear chains such as DNA or SDS-denatured proteins, for

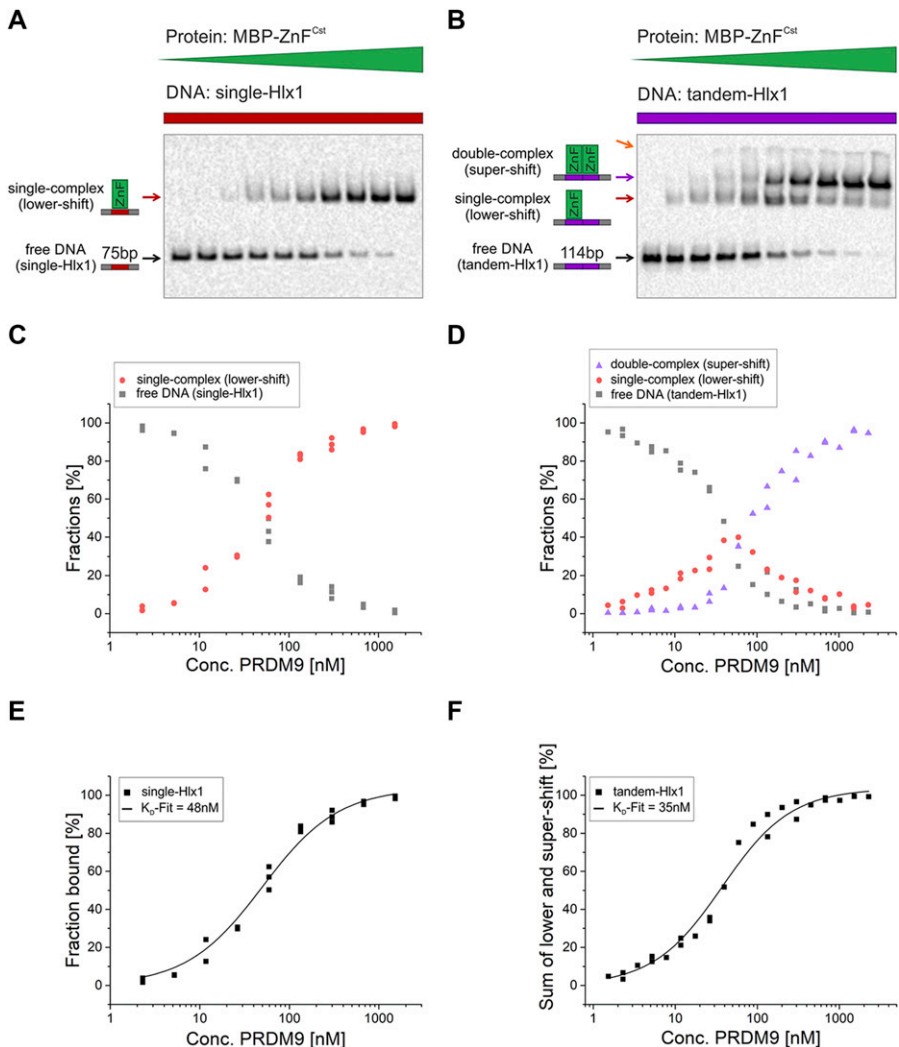

**Figure 1. Binding of the PRDM9–ZnF to one or two consecutive target sites.**

**(A, B)** Shown are titration EMSA experiments in which serial dilutions of MBP-ZnF^Cst (1.5 μM–2.3 nM for single-Hlx1; 2.3 μM–1.5 nM for tandem-Hlx1) were incubated with constant amounts of labelled target DNA (5 nM). Two different DNA targets were used, (A) single-Hlx1 with a length of 75 bp and (B) tandem-Hlx1 with a length of 114 bp, the latter carrying two consecutive *Hlx1*-binding sites. The lowest band (black arrow) is the unbound, free DNA and the shifted bands are the complex with either one (red arrow) or two (purple arrow) proteins at the target DNA, labelled as single complex (lower shift) and double complex (supershift), respectively. Pixel intensities of the unbound and shifted bands were quantified using the Image Lab software (Bio-Rad). Orange arrow indicates the wells of the EMSA gel giving slight signals of labelled DNA likely bound to big unspecific protein agglomerates. **(C, D)** Different fractions (% fraction) of the binding reaction (fraction unbound = free DNA, grey; lower shift indicating the single complex fraction after binding of one PRDM9 complex, red; and the supershift fraction indicating the double complex formation of two PRDM9 complexes bound to DNA, purple) were plotted against the PRDM9 concentration at a semilogarithmic scale with OriginPro8.5 software (OriginLab). **(E, F)** The fraction bound [FB = shift/(shift + unbound) × 100] was plotted against the PRDM9 concentration in a semilogarithmic scale and a $K_D$-fit was performed using a function for receptor–ligand binding in solution (as was described in Striedner et al (2017)). The $K_D$ for the (E) single-Hlx1 and (F) tandem-Hlx1 (sum of lower- and supershift) was estimated to be 48 and 35 nM, respectively. DNA concentrations and the resulting fractions bound of multiple experiments are listed in Table S1.

which the MW is inversely proportional to the logarithm of the migration distance in a gel (Ferguson, 1964; Lerman & Frisch, 1982; Slater & Noolandi, 1989). The migration of these linear, negatively charged chains is independent of the total charge and conformation of the molecule and follows the "reptation principle." This model proposes that the negative charge on one end of the molecule is sufficient to drive the rest of the molecule that migrates snakelike through the pores of the gel, oriented by the negative charge on one end and pulling the rest of the molecule through the same path (de Gennes, 1971, Lerman & Frisch, 1982; Lumpkin et al, 1985; Slater & Noolandi, 1989; Viovy, 2000) (for more details, see the Supplementary_Notes section of the Supplementary Information).

We developed two different strategies, *assay I* and *II*, to infer the MW of the DNA–PRDM9 complex in a polyacrylamide gel under nondenaturing conditions. As before, we used EMSA for visualizing the mobility of the complex and further estimate the protein stoichiometry by comparison with a standard series. In both assays, the migration of the complex was driven by the reptation of the long linear DNA overhangs flanking the complex. Note that under the used electrophoresis conditions, all the protein constructs are

positively charged and do not migrate into the gel unless they are bound to the DNA.

In the more conservative, but less accurate *assay I*, our standards were determined by the PRDM9–ZnF complexes (ZnF + DNA) with a constant conformation charge, but different MWs given by the length of the flanking DNA. Previous methods used a similar strategy of constant charge and conformation to derive a function of relative migration distance versus MW in a Ferguson plot (Ferguson, 1964; Hope & Struhl, 1987; Orchard and May 1993). For this purpose, we used in *assay I* the tandem-Hlx1 (carrying one or two complexes, as described in the previous section). Specifically, we designed DNA fragments of different lengths with one binding site (single-Hlx1) or two consecutive (tandem-Hlx1) binding sites, all with increasing nonspecific flanking sites (Figs 2A and S1), resulting in a lower shift (red rectangles) for the single-Hlx1 or lower- and supershift (purple rectangles, see Fig 2B) bands for the tandem-Hlx1 sequences. The lower shifts (single complex) were used as standards to infer the MW of the second complex in the supershifts (Figs 2B and S3). The standard curve with nine measurements resulted in a very high correlation of a linear regression function (all >97%;

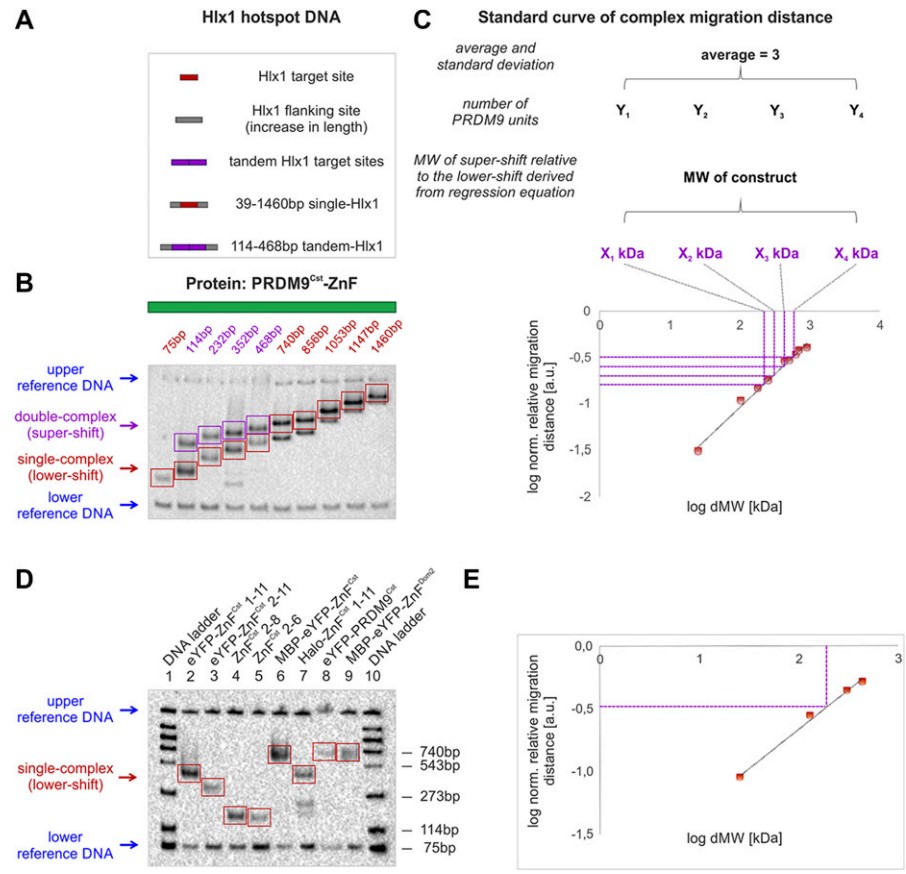

**Figure 2. Two strategies to infer the MW of PRDM9 from native gel electrophoresis.**
**(A)** Different sizes of biotinylated DNA containing one (red) or two (purple) *Hlx1*-binding sites (34-bp minimal target site for PRDM9$^{Cst}$) were used as DNA standards. The DNA fragments increase in nonspecific flanking sites (grey). **(B)** *Assay I*: DNA carrying one or two protein complexes was separated by a native polyacrylamide gel resulting in lower- and supershift bands (red and purple arrows/rectangles, respectively). Blue arrows indicate long- (4,368 bp) and short (220 bp) reference DNA, tested not to interact with PRDM9, but used to normalize the migration distance in each lane. Note that for high MW fragments, the free DNA shows up also on the gel but was not used for the analysis. **(C)** The migration distance of the PRDM9–DNA single complexes (lower shift), relative to the complex in the first lane (75 bp single-Hlx1) was plotted against the known relative increase in MW (dMW) between DNA targets in a log scale. The difference in migration distance of the supershift (double complex) relative to the lower shift (single complex) of four tandem-Hlx1 fragments was used to (1) estimate the MW representing the second protein complex using the regression equation ($X_1$–$X_4$); (2) to calculate the number of PRDM9 units based on the MW of the PRDM9 construct ($Y_1$–$Y_4$); and (3) to determine the average and SD of the units from the four tested tandem fragments. Note that complexes with lower MW get resolved better in electrophoresis and the estimation of the MW from the migration distance is more accurate. **(D)** *Assay II*: Binding complexes of eight different PRDM9 constructs with single-Hlx1 75 bp for PRDM9$^{Cst}$ constructs and single-Pbx1 75 bp for PRDM9$^{Dom2}$ constructs (lower shifts, red arrow/rectangles) were separated on the native EMSA gel. Lane 1 and 10 show a DNA ladder, with the respective fragment lengths shown on the right. Each lane included a lower (75 bp) and upper (75 bp, loaded 10 min before termination of electrophoresis) reference DNA (blue arrows) used to normalize the migration distance within each lane. The measurements were performed in four replicates of independent experiments. **(E)** The normalized migration distance of the DNA ladder bands in lane 1 and 10 relative to the shortest, 75 bp, molecule was plotted against the relative increase in MW in a log scale. The resulting regression equation was used to calculate the MW of the lower shift complexes and the number of protein units within the complex were estimated as described in panel C. Note that all DNA and protein concentrations used and EMSA conditions can be found in the Materials and Methods section and Supplemental Data 1.

Table S2) plotted in a log scale (Fig 2C and Table S2). The MW of the protein constructs was then estimated from the derived regression function as the average of four independent measurements (supershift) within one experiment (Fig 2B and C and Table S2).

In the simplified *assay II*, the MW of the different protein constructs was inferred by comparing the migration of the complex directly with free DNA standards (Fig 2D and E and Table S3). To further validate this strategy, we assessed PRDM9 constructs with different charges and conformations by adding different tags and PRDM9 domains, originating from the PRDM9$^{Cst}$ and PRDM9$^{Dom2}$ variants (Fig 3A). By comparing the migration of the shifted bands (lower shifts, red rectangles) relative to the migration of a DNA ladder (free DNA of different sizes), we estimated the MW of the PRDM9 complex and derived the protein units within each construct (Figs 3B and S4 and Tables S3 and S4). We compared the two developed assays by testing four ZnF$^{Cst}$ constructs with both methods. We did not observe differences in the estimated protein stoichiometry (Tables 1 and S4). This indicates that the migration of the complex in the native gel is driven invariably by the reptation of the long flanking DNA chain, independent of protein charge or conformation.

## The stoichiometry of PRDM9 is estimated to three units

To assess the protein stoichiometry of PRDM9 and the PRDM9 domain mediating this multimerization, we designed 11 different protein constructs missing selected domains of the PRDM9$^{Cst}$ and PRDM9$^{Dom2}$ (Fig 3A). In addition, constructs carried different tags such as enhanced yellow fluorescent protein (eYFP), maltose-binding protein (MBP), or Halo (HaloTag) produced by distinct expression systems, such as cell-free in vitro expression (IVE) and bacterial expression, or protein lysate protocols (Table S4 and Supplementary_Methods section of the Supplementary Information). Most bacterially expressed constructs were used as crude lysates without further purification obtained from the whole-cell (WC) fraction with cell debris or from the soluble fraction (SN) excluding cell debris. Only the lysate preparation for the construct containing the HaloTag included a purification step based on ion-exchange chromatography by using SP Sepharose, which rendered a semipure construct (for details see Supplementary_Methods section of the Supplementary Information).

Our results show that the MW for the complex inferred from EMSAs varies around three units (Fig 3B and Table 1, Fig S4 and

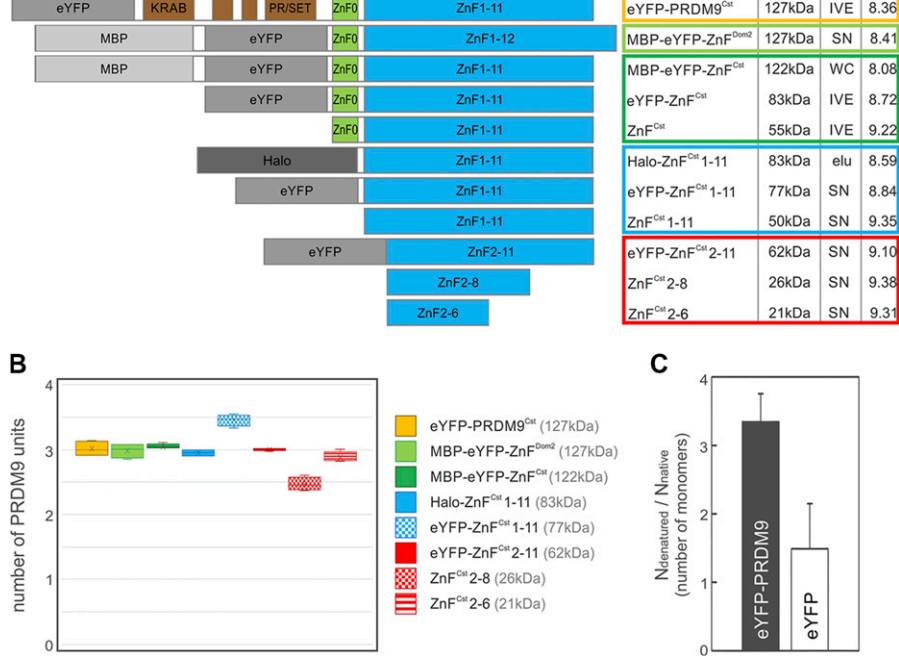

**Figure 3. PRDM9 multimerization is mediated within the ZnF array.**
**(A)** The different PRDM9 constructs used to infer the multimerization of PRDM9 are represented here. Domains of PRDM9 are colour-coded and additional tags are shaded in grey. Construct name, size, expression system (lys), and theoretical pI are shown on the right in a table format. Cell-free IVE; bacterially expressed WC fraction, WC; bacterially expressed SN, SN; semipure elution via ion-exchange chromatography, elu. **(B)** Box plot of the tested PRDM9 constructs representing the distribution of measured PRDM9 units within a multimer complex of *assay II*. Different PRDM9 constructs are colour-coded: yellow, full-length PRDM9$^{Cst}$; light green, ZnF domain of PRDM9$^{Dom2}$; dark green, ZnF domain of PRDM9$^{Cst}$; blue, tandem ZnF array of PRDM9$^{Cst}$ without ZnF0; and red, truncated ZnF array of PRDM9$^{Cst}$. **(C)** FCS of eYFP-labelled PRDM9 (Halo-eYFP-ZnF$^{Cst}$ 1–11) was used to estimate the concentration and mobility of fluorescent particles within a focal volume (see Table 2 and Fig S6, the Materials and Methods section, and Table S5). First, we obtained the number of particles ($N_{native}$) of fluorescent PRDM9 per focal volume in 1× TKZN buffer and compared it with the number of particles in 1× TKZN + 3 M urea ($N_{denatured}$). This concentration of urea dissolves the PRDM9 oligomer into monomers increasing the number of fluorescent particles per focal volume. We then estimated the number of PRDM9 monomers per oligomer in solution as the ratio of $N_{denatured}/N_{native}$

plotted here (the full data are shown in Tables 2 and S5). In comparison, the control eYFP (without PRDM9), which is commonly known as a monomer, did not change its number of particles with the addition of urea. Measurements were conducted at room temperature in DNA-binding buffer (10 mM Tris, 50 mM KCl, 50 μM ZnCl$_2$, and 0.05% NP-40, pH 7.5).

Table S4). Some constructs are off this average of three (e.g., eYFP-ZnF$^{Cst}$ 1–11 and ZnF$^{Cst}$ 2–8), which are likely experimental fluctuations given that a different tag, expression system, or assay rendered repeatedly a stoichiometric estimate of three for the same ZnF array. Note that the estimates from *assay II* are more accurate, given that the slower migration distances in *assay I* are prone to larger deviations due to the inverse exponential correlation of migration distance with MW (Fig S5).

We analysed the data from *assay I* and *II* independently with an ANOVA to assess whether the differences between the PRDM9 constructs can be explained by other than experimental/random variation (detailed analysis can be found in the Materials and Methods section and Supplementary_Statistical_Analysis section of the Supplementary Information). We show that the differences in unit sizes can neither be explained by construct size, additional

tags, and expression system, nor theoretical isoelectric point (Figs 3B and S5 and Table S4). Thus, given that most of the measurements average to three units per PRDM9 complex, especially the more accurate measurements (*assay II*), we conclude that PRDM9 is a trimer.

We compared the full-length PRDM9 (YFP-PRDM9$^{Cst}$; 127 kD) carrying all four different domains (KRAB, SSXRD, PR/SET, and ZnF) with a series of shortened constructs containing only a few ZnFs (Figs 3B and S5 and Table S4). All of the tested constructs were estimated to form a trimer. This also includes constructs expressing only the ZnF domain of two different murine alleles, PRDM9$^{Cst}$ and PRDM9$^{Dom2}$ without the KRAB, SSXRD, and PR/SET domains (MBP-eYFP-ZnF$^{Dom2}$, MBP-eYFP-ZnF$^{Cst}$, eYFP-ZnF$^{Cst}$, and ZnF$^{Cst}$). This suggests that the other three domains (KRAB, SSXRD, and PR/SET) of PRDM9 are not necessary for the multimer

**Table 1. Multimerization measured by *assay I* and *II*.**

| PRDM9 construct | Construct name | MW (kD) | Expression system | pI | Protein stoichiometry (CI) | |
|---|---|---|---|---|---|---|
| | | | | | *Assay I* | *Assay II* |
| Truncated ZnF array | eYFP-ZnF$^{Cst}$ 1–11 | 77 | bact. SN, WC | 8.84 | 3.8 (0.46) | 3.5 (0.09) |
| | eYFP-ZnF$^{Cst}$ 2–11 | 62 | bact. SN | 9.1 | 2.7 (0.44) | 3.0 (0.01) |
| | ZnF$^{Cst}$ 2–8 | 26 | bact. SN | 9.38 | 2.7 (0.46) | 2.5 (0.11) |
| | ZnF$^{Cst}$ 2–6 | 21 | bact. SN | 9.31 | 2.9 (0.30) | 2.9 (0.08) |

Four different PRDM9 truncated ZnF constructs measured in both *assay I* and *II* resulted in comparable average estimates of protein stoichiometry. The confidence intervals are given in parenthesis. The size of each construct (MW in kilodalton), the used expression system (bact. SN, SN of bacterial expression; WC, WC fraction including cell debris of bacterial expression), and the theoretical isoelectric point (pI) are shown.

formation and that the ZnF domain is sufficient to induce the multimerization.

We further removed individual ZnFs from the ZnF domain starting with ZnF0 (spaced 102 amino acids from the tandem array ZnF 1–11) of PRDM9$^{Cst}$ (Halo-ZnF$^{Cst}$ 1–11, eYFP-ZnF$^{Cst}$ 1–11, and ZnF$^{Cst}$ 1–11), eYFP-ZnF$^{Cst}$ 2–11 (missing ZnF0 and ZnF1), ZnF$^{Cst}$ 2–8, and ZnF$^{Cst}$ 2–6. Interestingly, even the smallest ZnF$^{Cst}$ 2–6 construct (with only 5 out of 11 ZnFs of PRDM9$^{Cst}$) bound as a trimer with the DNA. This strongly suggests that the trimer formation of active PRDM9 is mediated within the variable DNA-binding ZnF array and at least 5 of 11 fingers are sufficient to form a functional DNA-binding multimer. Moreover, we demonstrated that the PRDM9 trimerization is not dependent on the PRDM9 allele because both PRDM9$^{Cst}$ and PRDM9$^{Dom2}$ showed the same protein stoichiometry.

We also assessed the stoichiometry of PRDM9 in its free, soluble form. For this purpose, we used FCS to estimate the number of fluorescent molecules in a labelled, semipurified PRDM9 construct (Halo-eYFP-ZnF$^{Cst}$ 1–11). FCS counts the number of fluorescent molecules $N$ within a focal volume, estimated as an amplitude value (equivalent to $1/G(\tau)$ at $\tau = 0$) from FCS correlation curves (see Fig S6 and the Materials and Methods section; Tables 2 and S5). We compared the number of fluorescent particles of the labelled protein in native 1× TKZN buffer ($N_{native}$) and denaturing conditions of 3 M urea ($N_{denatured}$). Urea titration experiments showed that 3 M urea is enough to disrupt PRDM9 oligomers, but low enough to ensure eYFP emission (data not shown). This enabled us to calculate the number of PRDM9 monomers per oligomer in solution by the ratio $N_{denatured}/N_{native}$. We repeated this experiment with an equivalent eYFP construct, which forms mainly a monomer and thus served as a control with an expected ratio $N_{denatured}/N_{native}$ close to one. The ratio with and without 3 M urea for the eYFP-ZnF was estimated as 3.34 ± 0.41; n = 5, whereas for the eYFP, was estimated as 1.49 ± 0.66; n = 3 (see Fig 3C; Tables 2 and S5). Note that in both cases, the number of fluorescent monomers was slightly overestimated likely because of the nonidentical eYFP fluorescence in native versus denaturing conditions. However, the FCS results are congruent with the stoichiometry estimated with EMSAs and support the trimeric nature of PRDM9. Moreover, these data also suggest that PRDM9 does not change its stoichiometry upon DNA binding, and we conclude that PRDM9 is self-binding also in the

absence of DNA. This also agrees with a previous study reporting that DNA digestion with benzonase did not affect the multimerization between the ZnF arrays (Altemose et al, 2017).

We further tested in an additional experiment, if PRDM9–ZnF monomers are exchanged freely in a soluble, functional PRDM9 fraction (see Fig S7) before being complexed to DNA. For this purpose, we co-expressed two different variants of PRDM9–ZnF with the same number of ZnFs, but with or without eYFP tag (eYFP-ZnF$^{Cst}$ 2–11 and ZnF$^{Cst}$ 2–11 with 62 and 37 kD, respectively), and thus different MW. These two constructs, carried by two different plasmid vectors, were co-transfected in equimolar amounts in our bacterial expression system. We observed that both constructs were expressed at similar concentrations verified by a Western blot (Fig S7A). We observed two shifts with this co-expressed PRDM9–ZnF mixture in EMSA, each shift equivalent to the extract expressing only one of the two PRDM9–ZnF constructs (carried out in parallel). Intermediate complexes (mixture of short- and long protein versions) were not observed (Fig S7B). In contrast, co-immunoprecipitation (Co-IP) studies analysing multimerization reported a mixture of constructs within the complex (Baker et al, 2015b; Altemose et al, 2017). It is possible that these Co-IP studies enrich for the fraction that forms a mixture (heteromers); however, in our case, most soluble PRDM9 formed a homomer.

## Mass spectrometry verifies that the peptides in the shift are mainly derived from PRDM9

We used mass spectrometry to assess the composition of the protein in the shift (representing the PRDM9–DNA complex). This is an important aspect because our calculations of the stoichiometry assumed that only PRDM9 and DNA are the components in the shift. To confirm that the protein components of the complex come mainly from PRDM9, we isolated the complex of the semipure Halo-ZnF$^{Cst}$ 1–11 protein and the single-Hlx1 75-bp DNA from a native gel (Fig S8A) and analysed it by MALDI-TOF mass spectrometry (Fig S8B). We have extensively demonstrated that the only parameter affecting the mobility of the shift in our system is the MW of the complex, and none of the tags, purification level (crude extract, fraction in the supernatant, or purification level), or additives (e.g., polydIdC; nonspecific competitor in most EMSA reactions) affect the

**Table 2. PRDM9 stoichiometry inferred based on FCS measurements.**

| | Replicates | $N_{native}$ | $N_{denatured}$ | $N_{denatured}/N_{native}$ | Average (CI) |
|---|---|---|---|---|---|
| Halo-eYFP-ZnF$^{Cst}$ 1–11 | 1 | 0.258 | 0.990 | 3.84 | |
| | 2 | 0.242 | 0.829 | 3.43 | |
| | 3 | 0.295 | 0.988 | 3.35 | 3.34 (0.41) |
| | 4 | 0.419 | 1.237 | 2.95 | |
| | 5 | 0.356 | 1.123 | 3.15 | |
| eYFP | 1 | 0.228 | 0.336 | 1.47 | |
| | 2 | 0.235 | 0.289 | 1.23 | 1.49 (0.66) |
| | 3 | 0.229 | 0.404 | 1.76 | |

The free, soluble PRDM9 construct (Halo-eYFP-ZnF$^{Cst}$ 1-11) was purified via ion-exchange chromatography and then measured by FCS in 1× TKZN buffer and 1× TKZN buffer + 3 M urea. $N_{native}$ and $N_{denatured}$ represents the brightness of single fluorescent particles in the focal volume without and with urea, respectively, estimated as described in Equation (1) in the Materials and Methods section.

mobility in EMSAs. Thus, this complex isolated from a shift in a gel for mass spectrometry analysis should represent any of the other shifts, regardless of the construct used.

The mass spectrometric data of the Halo-ZnF$^{Cst}$ 1–11 showed that indeed the complex was formed mainly by PRDM9. First, none of the peptides identified by mass spectrometry was of bacterial origin based on searches of the NCBI or SwissProt databases (contamination with bacterial proteins could be likely, given that we used a bacterial expression system). Next, we measured the monoisotopic mass-to-charge (m/z) value of the peptide mass fingerprint (PMF) spectra and compared it with the theoretical m/z values of Halo-ZnF$^{Cst}$ 1–11 using ProteinProspector (University of California, prospector.ucsf.edu/prospector/mshome.htm). The 34 expected peptide ions from the expressed protein were detected in the PMF spectrum (Table S6). The software tool MS-Fit could assign 18 m/z values to peptides correlated with Halo-ZnF$^{Cst}$ 1–11; the other 7 m/z values were identified manually covering in total ~60% of the sequence of our construct. We also identified manually four peptides resulting from the autodigestion of trypsin and one from the used matrix (CHCA-cluster). We could not assign four further m/z values (maybe derived potentially from contamination).

To further confirm the PMF data, we analysed the MS/MS spectra of four prominent m/z values (1,338.61, 1,767.84, 1,810.76, and 1,908.01) (Fig S8C and Table S7). The MS/MS spectra analysis was performed by comparing the measured m/z values with calculated values of the corresponding amino acid sequences: for example, SFIASEISSIER has an m/z = 1,338.61, HQRTHTGEKPYVCR has an m/z = 1,767.84, SDKPDLGYFFDDHVR has an m/z = 1,810.76, and LLFWGTPGVLIPPAEAAR has an m/z 1,908.01. All four peptides belong to the amino acid sequence of the Halo-ZnF$^{Cst}$ 1–11. Note that we also detected persistent y-ion series in all four MS/MS spectra, as well as, matching b and a ions. The mass lists showing the matched PMF and MS/MS data are included in the Tables S6 and S7 and Fig S8.

### PRDM9 complex binds only one DNA molecule at a time

Because PRDM9 forms a multimer, we also asked whether the different ZnF arrays within the trimer can interact with more than one DNA molecule. The results of the tandem-Hlx1 experiment

described initially showed that in the presence of multiple binding sites, two independent complexes are formed on each binding site, suggesting that only one of the multiple ZnF arrays within the trimer interacts with the DNA-binding site. However, it is possible that the simultaneous interaction of multiple ZnF arrays within two adjacent target sites might have posed a physical constraint. Thus, we performed an additional test to assess if the multiple ZnF arrays within the trimer can interact simultaneously with more than one DNA molecule.

This time, we designed an experiment in which PRDM9 was incubated with a short and a long DNA sequence (75 and 273 bp) with the same binding site, but nonspecific flanking regions of different sizes. Each DNA–protein complex formed a unique shift in EMSAs, given the difference in MW of the DNA. In model 1, the trimer binds only one DNA molecule (either the long or the short DNA), and we expect two shifts in addition to the two free DNA sequences. Alternatively, in model 2, the trimer binds two or more DNA sequences, and we expect five bands: three shifts and two free DNA sequences are shown in Fig 4A. Our results clearly show the formation of a DNA–protein complex with either the short or the long DNA, but not both, demonstrating that only one of the three ZnF arrays in the multimer actively binds to the DNA (Fig 4B). We further extended this experiment in Fig S9 and tested multiple long- and short DNA fragment combinations (75 + 189, 189 + 856, and 271 + 543). Note that with our EMSAs, we can resolve complexes on fragments ranging from 75 to 1,460 bp differing in size by ~100 bp-steps (see almost perfect correlation of migration distance with MW in Fig 2C). We did not observe intermediate-sized bands in any of the tested combinations, but we exclusively counted only two shifts (one with the short DNA and one with the long DNA). It could be argued that the nonsaturation conditions of the protein could explain this binding behaviour because the protein concentration was ~10-fold higher than that of the DNA. However, in an experiment in which a long DNA (75 bp) was incubated with increasing concentrations of a short DNA (39 bp) reaching an excess of DNA over protein, we still do not observe additional shifts (see Fig S10; data are from Striedner et al (2017), Tiemann-Boege et al (2017)). In this latter experiment, we observed only one shift because only the long DNA was biotinylated, but the shorter DNA in excess (titrated from 0 to 1,500 nM) was not, to avoid smearing in the EMSA. Based on these results, we propose a

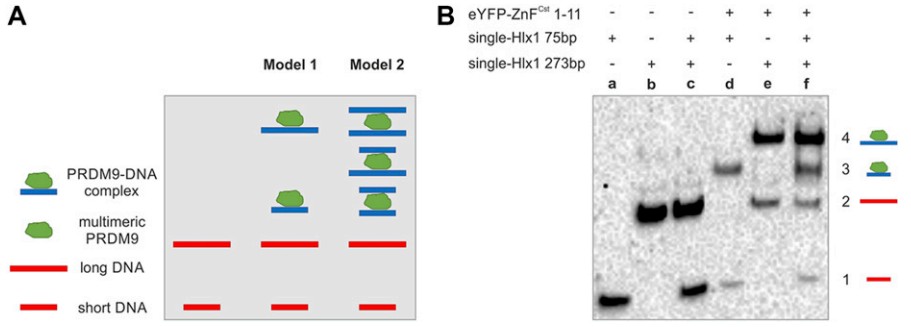

**Figure 4. PRDM9 complex forms with only one target molecule.**
**(A)** The two models represent the binding of the multimeric PRDM9 complex (green) to a short- and long DNA containing the same binding site. The final MW of the protein–DNA complexes varies, resulting in distinct migration distances in the EMSA gel. When mixing equimolar amounts of short- and long DNA with PRDM9, the protein will randomly bind either the short or the long DNA. Model 1 represents the banding pattern if the protein complex binds only to one DNA molecule at a time resulting in two shifts and four different bands in total: (1) short DNA, (2) long DNA, (3) protein + short DNA, and (4) protein + long DNA. Model 2 shows the banding pattern if the multimeric protein binds two DNA molecules at a time, resulting in five different bands: (1) short DNA, (2) long DNA, (3) protein + 2× short DNA, (4) protein + 1× short DNA + 1× long DNA, (5) protein + 2× long DNA. **(B)** The EMSA was performed with eYFP-ZnF$^{Cst}$1–11 (0.25 μl) mixed with two *Hlx1* DNA fragments of size 75 and 273 bp at equal molarities (5 nM). The experiment was repeated using more combinations of different short- and long DNA sequences (75, 189, 273, 543, and 856 bp) and a different protein construct with the same number of ZnF repeats (Halo-ZnF$^{Cst}$ 1–11) as shown in Fig S9.

binding model in which only one of the ZnF arrays interacts with the DNA. The remaining two ZnF arrays could be involved in protein–protein interactions stabilizing the multimer.

# Discussion

## PRDM9 forms a trimer

In this work, we developed an approach to infer the stoichiometry of PRDM9 using a combination of techniques such as native gel electrophoresis (specifically EMSAs), mass spectrometry, and FCS. All these techniques support the trimeric nature of PRDM9. Our data demonstrate that multimerization is independent of the tested PRDM9 allele (PRDM9$^{Cst}$ and PRDM9$^{Dom2}$). Neither functional tags such as eYFP, MBP, and Halo, nor expression systems (bacterial or IVE), or protein purity did influence the protein stoichiometry in all 11 tested protein constructs, confirming that our approach is very robust. By fingerprinting the PRDM9–DNA complex with mass spectrometry, we confirmed that the complex contains mainly PRDM9 peptides. With FCS, we inferred that the free, soluble PRDM9–ZnF is also formed by three units, which is congruent with the estimated trimeric structure of the complex (PRDM9 + DNA), estimated using migration distances in native gel electrophoresis. Thus, we conclude that the stoichiometry of PRDM9 of a trimer is already adapted as a free soluble protein and is not catalysed by the binding to DNA.

Previous studies also reported that PRDM9 forms functional multimeric complexes with at least two or more units (Baker et al, 2015b; Altemose et al, 2017). This was shown in cells co-transfected with PRDM9 alleles carrying different tags and/or ZnF domain-binding properties. Co-IP experiments targeting only one tag showed that the precipitate contained both tags, suggesting multimerization (Baker et al, 2015b; Altemose et al, 2017). In contrast to these Co-IP studies, we did not observe complexes combining both sizes (mixture of isomers) when co-expressing two PRDM9 constructs of different MWs (the result of an additional eYFP tag). An explanation compatible with both opposing results could relate to the possibility of mixtures organized mainly in larger agglomerates, which are not detectable in EMSAs (albeit very large scattered particle sizes were occasionally observed in FCS), but might be specifically enriched by Co-IP. Our co-expression results do not preclude multimerization, but rather can be interpreted as the preferential formation of isomers and the lack of interchange between monomers once multimerized.

Note that also in Co-IP experiments, it was observed that PRDM9 prefers to interact with the same variant rather than forming heteromeric complexes. In an experiment co-expressing three different PRDM9 variants (chimp-V5, chimp-HA, and human-HA), twice as many isomers (chimp/chimp or human/human) were formed as heteromers (Altemose et al, 2017). Similarly, no evidence for heteromer formation was observed in vivo by the trans-complement methyltransferase activity in mice heterozygous for PRDM9$^{Dom2}$ and PRDM9$^{Cst-YF}$ (PRDM9$^{Cst}$ variant with a methyl-transferase knockout mutation) (Diagouraga et al, 2018). This suggests that self-binding of PRDM9 could be quite stable with little

exchange between monomers once assembled within a cell. The question is whether preferential formation of a homo/heteromer also depends on ZnF divergence.

## PRDM9 multimerization is mediated within the ZnF domain

We removed the KRAB, SSXRD, PR/SET domain, the single ZnF0, and even shortened stepwise the PRDM9–ZnF array to the smallest construct with only 5 of 11 ZnFs. In none of these constructs, we observed a change in stoichiometry. Thus, we conclude that the PRDM9 multimerization is mediated within the variable DNA-binding ZnF domain. Moreover, 5 of 11 fingers within the PRDM9$^{Cst}$ ZnF domain, more precisely ZnFs 2–6, are sufficient for the formation of a stable trimer that binds specifically to DNA. This is also congruent with a study using co-IP experiments of different co-expressed PRDM9 constructs reporting that PRDM9–PRDM9 interactions occur within the ZnF domain, albeit weak self-binding interactions in the absence of the ZnF array were also observed (Altemose et al, 2017).

There are several reports of multimerization in other ZnF proteins that can either form both homo- and heteromers; although homomers are preferred (Iuchi, 2001; Mccarty et al, 2003). In these studies, complexes formed by maC2H2 type ZnFs (with several consecutive ZnFs) were interpreted as "higher order structures" or dimers using yeast two-hybrid systems, co-transfection of isoforms and gel shift assays, but no exact stoichiometry was established (Sun et al, 1996; Tsai & Reed, 1998). Multimerization is usually mediated via ZnFs not participating in DNA recognition using two different modes: hydrophobic interactions through the ZnF surface (Wang et al, 2001), as it was shown for proteins like GL1 (Pavletich & Pabo, 1993) and SW15 (Dutnall et al, 1996), or ZnF–ZnF interactions mediated by the same amino acids conferring the DNA sequence specificity (Wolfe et al, 2000; Mccarty et al, 2003). It has been argued that this multimerization serves to increase the binding affinity and efficiency to target DNA sequences (reviewed in Iuchi (2001)).

PRDM9 has many parallels to the maC2H2 Zn finger subfamily, which have multiple adjacent ZnFs (four or more) within a row, such as TFIIIA, Ikaros, or Roaz (reviewed in Iuchi (2001)). Only a subset of about 24–75% of maC2H2 ZnFs are part of the DNA sequence rec-ognition, whereas the rest is free for other roles such as RNA or protein–protein interactions (reviewed in Iuchi (2001)). Interest-ingly, those ZnFs of the maC2H2 family, which are not participating in DNA binding, often mediate dimerization, which can also in-crease the binding affinity, as it was observed for Ikaros (Sun et al, 1996) and Roaz (Tsai & Reed, 1998).

Similar to the other maC2H2 members, not all the ZnFs within the array of PRDM9 are necessary to form a stable and sequence-specific binding with DNA (Altemose et al, 2017; Striedner et al, 2017). Our truncation product ZnF2–6 of PRDM9$^{Cst}$ showed that five ZnFs are sufficient to form a multimer. However, we also know that all these ZnFs are also in direct contact with the DNA because at least five ZnFs are required for a stable and sequence-specific DNA binding, as was shown in several instances (Altemose et al, 2017; Striedner et al, 2017); thus, it is likely that the ZnF array of PRDM9 catalyses the multimerization via hydrophobic interactions on the ZnF surface. However, we cannot exclude the possibility that in a longer PRDM9–ZnF array, some ZnFs are involved in a specific

ZnF–ZnF interaction. For example, for the human PRDM9[A], it was suggested, based on enrichment of DNA-binding motifs, that ZnFs 5 and 6 could act as linkers between up- and downstream ZnFs or might have other functions such as the ZnF–ZnF interaction (Altemose et al, 2017).

### The PRDM9 trimer binds only one DNA target at a time

Given the oligomeric nature of PRDM9, the question remains how many DNA molecules can be bound by the polymeric complex. We showed that the tandem-Hlx1 with two binding sites forms two independent complexes. Moreover, the PRDM9 trimer binds only to one DNA molecule in an equimolar mixture of long- and short DNA sequences. These data suggest that the multimer only binds one DNA target molecule at a time, even though three ZnF domains would be available.

It is possible that the two other ZnF domains are important in mediating a stable ZnF–ZnF interaction already established before the DNA binding, explaining why ZnF proteins are often found in multimers. However, we cannot exclude that the other domains of PRDM9 (e.g., KRAB, SSXRD, or PR/SET) engage independently within the multimer. This view was documented in a previous report, which described that different domains within the multimer are functionally active by the cotransfection of two human PRDM9 variants binding different DNA targets (PRDM9[A] and PRDM9[C] with a methyltransferase knockout mutation). H3K4me3 marks were enriched at hotspots recognized by the catalytically dead PRDM9[C], and thus must have been placed by the methyltransferase activity of PRDM9[A] as part of a multimer, albeit at a comparatively less strength and number (Baker et al, 2015b).

### PRDM9 multimerization in the context of hybrid sterility

*Prdm9* has been identified to play an important role in hybrid sterility (Mihola et al, 2009). Interestingly, only certain combinations of heterozygous *Prdm9* alleles are incompatible in a specific hybrid genetic background (Flachs et al, 2012, 2014). The process is not yet fully understood, but PRDM9 recognition sequences erode or change by processes like biased gene conversion or mutagenesis (Arbeithuber et al, 2015), and develop a weaker binding (reviewed in Tiemann-Boege et al (2017)). Thus, in crosses of two different murine species or subspecies, each of the two PRDM9 variants preferentially bind sequences of the non–self-chromosome located at different chromosomal positions. This leads to an asymmetric binding of PRDM9 and to an asymmetric distribution of DSBs between homologues, which are linked to chromosome asynapsis and hybrid sterility (Davies et al, 2016; Smagulova et al, 2016). Interestingly, only some murine crosses are sterile for reasons yet unknown. One explanation for *Prdm9* allele compatibility in heterozygous individuals is the combination of PRDM9 alleles within a certain genetic background resulting in the preferential hotspot usage by only one *Prdm9* allele. This dominance of hotspot activation by one allele was shown in heterozygous individuals in both humans for the C versus A allele (Pratto et al, 2014) and in mice for the 9R versus 13R allele (Brick et al, 2012) or the B6 versus CAST allele (Baker et al, 2015b; Smagulova et al, 2016).

What are the molecular mechanisms behind this dominance? We speculate that within a multimer, the stronger allele could direct the binding and sequester the weaker allele resulting in a hotspot landscape determined by the dominant PRDM9. Moreover, the activity of the weaker allele would be masked within a trimeric structure with most oligomers having at least one strong (dominant) unit that could be directing the binding.

A trimer would also have an interesting effect in terms of dosage within a heterozygous context because the two PRDM9 variants are physically coupled in a 2:1 ratio, but the trimer recognizes only one target molecule at a time. It has been shown in several instances that PRDM9 dosage plays a crucial role in fertility with both homo- and hemizygous *Prdm9* null mice showing complete or partial sterility because of a drastically reduced number of active hotspots (Hayashi et al, 2005; Baker et al, 2015a). PRDM9 dosage also determines the number and activity of hotspots. Removing or over-expressing a certain PRDM9 allele, and, therefore, increasing the PRDM9 dosage, could rescue fertility in sterile hybrid crosses (Flachs et al, 2012). Moreover, certain heterozygous F1 hybrids also show partial asynapsis with a strong bias towards the smallest autosomes, as it was shown for PWD × C57BL/6 crosses (*Prdm9[Msc/Dom2]*), which could be rescued by introducing a minimum of 27 Mb con-subspecific homologous sequence to one of the chromosome pairs restoring the symmetric hotspot distribution (Gregorova et al, 2018). This suggests that a certain number of active hotspot sites are required for successful meiotic progression, which among others is controlled by the dosage of PRDM9. It is possible that the variable formation of hetero-/homomers impacts PRDM9 dominance patters and potentially plays a role in the dosage sensitivity of PRDM9.

## Conclusions

Taken together, here we demonstrate using a series of multiple biophysical methods that PRDM9 forms a trimer as a free soluble protein, as well as, when complexed to the DNA. This trimerization is likely mediated by ZnF interactions within the long ZnF domain, and unexpectedly the trimer binds to only one specific DNA target molecule at a time. With the possibility that two PRDM9 variants form a trimer in a 2:1 ratio within a heterozygous organism, we provide important insights in the nature of the ZnF–DNA interaction and DNA targeting of PRDM9 in general, and also in the context of hybrid sterility because dominance and dosage likely correlate with PRDM9 homo- and heteromerization.

## Materials and Methods

### DNA sources

DNA fragments were produced via PCR amplification (Table S8) of genomic DNA of the B6 mouse using biotinylated or unmodified primers or hybridization of complementary single-stranded oligonucleotides (sequences are listed in Table S9). Details are shown in Supplementary_Methods section of the Supplementary Information.

## Cloning & expression of PRDM9 constructs

Distinct coding sequences of *Prdm9*[Cst] (CAST/EiJ strain, *M. musculus castaneus* origin) and *Prdm9*[Dom2] (C57BL/6J strain, *M. musculus domesticus* origin) were cloned into different vector systems for bacterial (pOPIN vector) and cell-free in vitro (pT7-IRES-MycN vector) expression as it was described by Striedner et al (2017). The inserts were prepared via specific PCR amplification and cloned into the desired vector using restriction enzyme-based cloning. The different constructs were designed to involve different tags such as His-tag, MBP, or eYFP as well as different parts of the *Prdm9*[Cst] or *Prdm9*[Dom2] coding region. Two PRDM9[Cst] constructs including a His$_6$-HaloTag (Halo), which was kindly provided by the Petkov Lab (Center for Genome Dynamics, the Jackson Laboratory, Bar Harbor, ME, USA) were used for bacterial expression. For most expressed proteins, a crude lysate was used for further experiments. Only Halo-PRDM9[Cst] ZnF1-11 and Halo-eYFP-PRDM9[Cst] ZnF1-11 were semipurified by ion-exchange chromatography based on a protocol described by Walker et al (2015) (Fig S11). A detailed description about cloning, expression, lysate preparation, and purification can be found in Supplementary_Methods section of the Supplementary Information. In summary, we used the following constructs:

Construct 1: His-eYFP-PRDM9[Cst] in pT7-IRES-MycN vector (IVE system).
Construct 2: His-MBP-eYFP-PRDM9[Dom2] ZnF in pOPIN-M vector (bacterial expression).
Construct 3: His-MBP-eYFP-PRDM9[Cst] ZnF in pOPIN-M vector (bacterial expression).
Construct 4: His-eYFP-PRDM9[Cst] ZnF in pT7-IRES-MycN vector (IVE system).
Construct 5: His-PRDM9[Cst] ZnF in pT7-IRES-MycN vector (IVE system).
Construct 6: His-Halo-PRDM9[Cst] ZnF1-11 in pH6HTN-His$_6$-HaloTag-T7 vector (bacterial expression).
Construct 7: His-eYFP-PRDM9[Cst] ZnF1-11 in pOPIN vector self-made (bacterial expression).
Construct 8: His-PRDM9[Cst] ZnF1-11 in pOPIN vector self-made (bacterial expression).
Construct 9: His-eYFP-PRDM9[Cst] ZnF2-11 in pOPIN vector self-made (bacterial expression).
Construct 10: His-PRDM9[Cst] ZnF2-8 in pOPIN vector self-made (bacterial expression).
Construct 11: His-PRDM9[Cst] ZnF2–6 in pOPIN vector self-made (bacterial expression).
Construct 12: His-MBP-PRDM9[Cst] ZnF in pOPIN-M vector (bacterial expression)—only used for protein titration experiments.
Construct 13: His-Halo-eYFP-PRDM9[Cst] ZnF1-11 in pH6HTN-His$_6$-HaloTag-T7 vector (bacterial expression)—only used for FCS measurements.

## EMSAs

### General EMSA protocol
Different EMSA experiments did vary in terms of binding reactions, incubation, and electrophoresis times but followed the general EMSA protocol described in Striedner et al (2017). All details about

EMSA experiments can be found in Supplementary_Methods section of the Supplementary Information and Table S10.

### Image analysis
Image analysis was performed using the Image Lab software 5.1.1 (Bio-Rad). The lanes and bands were defined manually, then the migration distances and pixel intensities could be quantified and analysed further using Excel and OriginPro software (Origin Lab).

## Inference of MW of the PRDM9–DNA complex from native gel electrophoresis

We analysed two different PRDM9 alleles (PRDM9[Cst], from the CAST/EiJ strain of *M. musculus castaneus* origin; and PRDM9[Dom2] from the C57BL/6J strain of *M. musculus domesticus* origin) targeting specifically the DNA of the *Hlx1* or the *Pbx1* hotspot, respectively (Billings et al, 2013). The PRDM9 protein was produced by bacterial or cell-free in vitro recombinant expression of different constructs carrying different tags, such as eYFP, MBP, His$_6$-HaloTag (Halo), or no tag. In addition, some of the domains of PRDM9 or repeats of the ZnF array were removed. In total, we tested 11 different protein constructs (for details, see Supplementary_Methods section of the Supplementary Information): eYFP-PRDM9[Cst], MBP-eYFP-ZnF[Cst], eYFP-ZnF[Cst], ZnF[Cst], Halo-ZnF[Cst] 1–11, eYFP-ZnF[Cst] 1–11, ZnF[Cst] 1–11, eYFP-ZnF[Cst] 2–11, ZnF[Cst] 2–8, ZnF[Cst] 2–6, and MBP-eYFP-ZnF[Dom2]. This large range of different sized protein constructs varied in conformation and charge; yet, rendered similar relative mobilities in EMSAs confirming that in our set-up, the migration of the complexes was mainly dependent of its MW.

### Assay I
For multimer *assay I*, we used the advantage of the tandem-Hlx1 molecules resulting in supershift bands representing a second PRDM9 complex bound. Each experiment was used to analyse only one type of PRDM9 construct. The protein was bound to six single-Hlx1 (75, 740, 856, 1,053, 1,147, and 1,460 bp) and four tandem-Hlx1 (114, 232, 352, and 468 bp) (see Fig S3A), or three single-Hlx1 (75, 543, and 740 bp) and two tandem-Hlx1 (114, 232 bp; for very small protein constructs, see Fig S3B), which increased in unspecific flanking sites. Protein–DNA–binding complexes were separated by the sieving effect of a native 5% polyacrylamide gel driven by the negative charges of the DNA resulting in lower shift (only one PRDM9 protein bound) or lower- and supershift (one or two PRDM9 proteins bound, respectively) bands. A long (4,368 bp, usDNA1) and short (220 bp, usDNA2) unspecific reference DNA were included, tested not to interact with the protein, which were then used to normalize for the migration distance of the different bands in each lane: ([usDNA2] – [usDNA1]) – ([lower shift] – [usDNA1]) = [lower shift]$_{norm}$. The relative increase in [lower shift]$_{norm}$ compared with lane 1 was plotted against the relative increase in molecular weight [dMW]$_{lower shift}$, which is given by the size of the DNA fragment, in a logarithmic scale resulting in a linear regression. [supershift]$_{norm}$ was then used to determine [dMW]$_{supershift}$ based on the regression function. [dMW]$_{supershift}$ – [dMW]$_{lower shift}$ = [dMW] for each tandem DNA sample represents one additional PRDM9 complex. By using the MW of the monomeric PRDM9 construct (e.g., 55 kD for ZnF[Cst]), the protein stoichiometry

(#PRDM9) can be calculated from (dMW). With four tandem-Hlx1 DNA fragments, four values for protein stoichiometry have been observed for each experiment. The experiments for one type of PRDM9 construct were replicated at least three times, except for the in vitro expressed constructs eYFP-ZnF$^{Cst}$ and ZnF$^{Cst}$. As a control for DNA impurities, unbound DNA fragments were separated and detected along the native gel without complexing with protein samples (see Fig S3C).

### Assay II

To evaluate the *multimer assay II* experiments, one EMSA was used to investigate eight different types of PRDM9 constructs which was replicated four times. All protein constructs were bound to a DNA fragment of 75 bp (for PRDM9$^{Cst}$ constructs, the *Hlx1* hotspot was used; for PRDM9$^{Dom2}$ constructs, the *Pbx1* hotspot was used). Unspecific reference DNA fragments of 75 bp (usDNA1, loaded 10 min before termination of electrophoresis; usDNA2, loaded at the beginning of electrophoresis) were included in each lane. The calculation of the PRDM9 stoichiometry was performed the same way as for *assay I*. However, a ladder of unbound DNA (75, 114, 273, 543, and 740 bp) was used as standards instead of the lower shift. Note that the highest DNA standard (740 bp) was chosen to be slightly higher than the highest shifted band to ensure a perfect and accurate relationship between MW and migration distance. The stoichiometry was derived from the migration distance of the lower shift band.

More details to calculate the protein stoichiometry using multimer *assay I* and *II* can be found in Tables S2 and S3.

## Statistical analysis

We tested for significant differences of calculated protein stoichiometry between different PRDM9 constructs for *assay I and II* separately using an ANOVA. A detailed description of the statistical analysis can be found in Supplementary_Statistical_Analysis section of the Supplementary Information.

## Mass spectrometry

### Chemicals

Acetone p.a., acetonitrile p.a. (ACN), acetic acid (96%), and ethanol (EtOH; 96%) were obtained from Merck. Alpha-cyano-4-hydroxycinnamic acid (CHCA), ammonium hydrogen carbonate ($NH_4HCO_3$), Coomassie brilliant blue R250 (CBB), DTT, iodoacetamide, and TFA were obtained from Sigma-Aldrich. 5% mini-PROTEAN TBE gel was obtained from Bio-Rad. Sequencing grade–modified trypsin was obtained from Promega and $C_{18}$ ZipTips from Merck Millipore.

### DNA preparation

A single-stranded DNA fragment was extended to produce the 75-bp fragment of the murine *Hlx1* hotspot. Therefore, 25 μM of the synthetic oligonucleotide ssHlx1-75b was hybridized with 25 μM of the primer single-Hlx1_R1 (sequences are listed in Table S9) in a 30-μl reaction by incubating for 5 min at 95°C and cooling down for 1 h. The hybridized DNA sample was supplemented with 1× NEB buffer 2.1 (NEB), 1 mM dNTPs (Biozym), and 6.75 units T4 DNA polymerase

(NEB) in a 56-μl reaction and incubated for 1 h at 12°C to start DNA extension. To remove the remaining single-stranded DNA fragments, the sample was digested with exonuclease I (NEB) as described in Supplementary_Methods section of the Supplementary Information. To purify the DNA, the sample was mixed with 2 μl Co-Precipitant Pink (VWR) and 0.5 volumes of 5 M $NH_4OAc$. Furthermore, two volumes of pure EtOH were added and mixed by inverting. For total DNA precipitation, the sample was incubated at –20°C for 30 min followed by centrifugation at maximum speed for 30 min at 4°C. The supernatant was discarded and the pellet washed with 1 ml 80% EtOH. After a final centrifugation step of 5 min at full speed and 4°C, the supernatant was carefully discarded and the pellet was dried at room temperature. The DNA sample was dissolved in 20 μl nuclease-free water (Sigma-Aldrich).

### Preparation of binding reaction

To prepare the PRDM9–DNA binding complex, 7 μl of semipure Halo-ZnF$^{Cst}$ 1–11 were mixed with 2 μM Hlx1-75 bp DNA in a 20-μl binding reaction supplemented by 1× binding buffer (1× TKZN = 10 mM Tris, 50 mM KCl, 0.05% NP-40, and 50 μM $ZnCl_2$) and incubated for 60 min at room temperature. The reaction was prepared twice.

### Gel electrophoresis

20-μl sample solution was supplemented by 1× DNA loading dye (Thermo Fisher Scientific) and applied onto the gel. Electrophoresis was performed on 5% mini-PROTEAN TBE (Bio-Rad), 10 wells in 30 μl gels, using 1× TBE (89 mM Tris, 89 mM boric acid, and 3 mM EDTA) as running buffer. Constant voltage was set to 100 V (50 mA/gel); after 40 min, the separation was stopped.

After gel electrophoresis, Coomassie staining with CBB R250 was performed. The gel was fixed (45% EtOH and 5% acetic acid in water) for 45 min and subsequently stained (0.1% CBB R250 in 45% EtOH and 5% acetic acid in water) for 1 h. Destaining was performed using two solutions: destain solution I (40% EtOH and 7% acetic acid in water) for 30 min, followed by destain solution II (5% EtOH and 7% acetic acid in water) overnight for clearing the background to obtain distinct protein bands.

### In-gel tryptic digestion

The protein gel band was excised and cut into small pieces. To remove contaminants and CBB R250 stain, various washing steps, each lasting 15 min, were applied: once with water, two times with ACN/water (1:1), once with 100% ACN, and once with ACN/50 mM $NH_4HCO_3$, pH 8.5 (1:1, vol/vol). Gel pieces were dried in a vacuum centrifuge. Subsequently, disulfide bridges were reduced with 100 mM DTT (15.4 mg/ml in 50 mM $NH_4HCO_3$ pH 8.5) for 45 min at 56°C and alkylated with 55 mM iodoacetamide (10.2 mg/ml in 50 mM $NH_4HCO_3$, pH 8.5) for 30 min at room temperature in the dark. Another washing step with ACN/50 mM $NH_4HCO_3$, pH 8.5 (1:1), was performed. Gel pieces were dried in the vacuum centrifuge. Subsequently, the gel pieces were incubated with 15 μl digestion solution (12.5 ng/μl trypsin in 50 mM $NH_4HCO_3$) for 15 min and then coated with 25 μl 50 mM $NH_4HCO_3$, pH 8.5. The protein was digested at 37°C overnight.

Peptide extraction from the gel pieces was performed with ACN/50 mM $NH_4HCO_3$, pH 8.5 (1:1), ACN/0.1% TFA (1:1), and 100% ACN, each

step lasting 15 min. The extracts were pooled and lyophilised in a vacuum centrifuge.

### MALDI sample preparation

First, the stainless steel MALDI target was prepared by application of 1 µl CHCA matrix solution (6 mg/ml in acetone). After evaporation of acetone at room temperature, a thin homogenous layer of matrix crystals was obtained.

Peptides were dissolved in 0.1% TFA and desalted using $C_{18}$ ZipTips. The tips were activated with ACN/0.1% TFA (1:1) and equilibrated with 0.1% TFA. After binding of the peptides, salts and detergents were removed by washing the tips five times with 0.1% TFA. Elution was performed using 1.5 µl ACN/0.1% TFA (6:4), which was directly applied onto the prepared CHCA layer on the MALDI target. The sample spot was dried at room temperature and subsequently transferred into the AXIMA Performance instrument.

### Instrumentation

Gel electrophoresis was performed on a Mini-PROTEAN (Bio-Rad) vertical electrophoresis cell connected to a Consort EV265 (VWR). MALDI-TOF spectra were acquired on an AXIMA Performance instrument (Shimadzu). The AXIMA Performance is equipped with a nitrogen laser ($\lambda$ = 337 nm) and it was operated in positive ion, reflectron mode using pulsed extraction. PMF mass spectra were acquired by averaging 500 and MS/MS spectra by averaging up to 2,500 unselected and consecutive laser shots. No smoothing algorithm was applied before data analysis.

### FCS

### Oligomeric state of PRDM9 in solution

FCS serves to measure the concentration (Horner et al, 2012; Hoomann et al, 2013; Knyazev et al, 2013; Erokhova et al, 2016) and mobility (Horner et al, 2009, 2013; Antonenko et al, 2012) of fluorescently labelled particles within a microscopic volume (<fl) element. We exploited FCS to determine the oligomeric state of PRDM9 with minor modifications of our previously described method (Horner et al, 2015, 2018; Horner & Pohl, 2018). In brief, recordings of temporal fluorescence intensity fluctuations *I* of eYFP-labelled PRDM9, using our commercial laser scanning microscope equipped with avalanche diodes (LSM 510 META Confocor 3; Carl Zeiss), allowed calculation of autocorrelation curves (see Fig S6):

$$G(\tau) = \frac{1}{\langle N \rangle} \left[ 1 + \frac{\tau}{\tau_d} \right]^{-1} \left[ 1 + \left( \frac{r_0}{z_0} \right)^2 \frac{\tau}{\tau_d} \right]^{-\frac{1}{2}}, \qquad (1)$$

with $\langle N \rangle$, $\tau_d$, $\tau$, and $z_0$ and $r_0$, which are the number of fluorescently labelled particles in the focal volume, their diffusion time through the focal volume, the lag time, and the extent of the focus in direction of the optical axis and perpendicular to it, respectively (see Table S5). $G(\tau)$ describes essentially the self-similarity of the fluorescence intensity fluctuations. For different $\tau$ values, this leads to the typical shape of an autocorrelation curve. The amplitude of $G(\tau)$ at $\tau$ = 0 is inversely proportional to $\langle N \rangle$ (= 1/G(0)) and, hence, can be used to calculate concentrations, whereas the turning point

of $G(\tau)$ denotes $\tau_d$, which is a measure of mobility and can be recalculated to diffusion constants. The overall fluorescence intensity $\langle I \rangle$ of a sample consists of the fluorescence intensities of single diffusing particles $I_{unit}$ times $\langle N \rangle$. We calibrated the system including $r_0$ and $z_0$ by using rhodamine-6G, which has a known diffusion coefficient of 426 $\mu m^2$/s (Saffman & Delbruck, 1975). First, we obtained the number $N_{native}$ (equivalent to 1/G(0)) of PRDM9 per focal volume in 1× TKZN buffer (10 mM Tris, 50 mM KCl, 50 µM $ZnCl_2$, and 0.05% NP-40, pH 7.5) using Equation (1). Subsequently, we dissolved PRDM9 oligomers by the addition of 3 M urea to estimate the number $N_{denatured}$ of single PRDM9 subunits. Denaturation of the PRDM9 complex leads to a multiplication of fluorescent particles in the focal volume due to the fact that one PRDM9 oligomer dissociates into several PRDM9 monomers. This enabled us to calculate the number of PRDM9 monomers per oligomer in solution by the ratio $N_{denatured}/N_{native}$ (see Table 2). We repeated this experiment with an equivalent eYFP construct (see Supplementary_Methods section of the Supplementary Information "Expression and semipurification of murine PRDM9 constructs and eYFP"), which served as a negative control assumed as a monomer, although they can also form weak dimers in solution (Shaner et al, 2005; Nakagawa et al, 2012). We also tested the addition of 3 M urea + 2% SDS + 5.3% 2-mercaptoethanol (data not shown), but this stronger denaturation resulted in bizarre data also in the eYFP control likely because of compromised eYFP emission. We also measured with FCS the eYFP-labelled PRDM9 in the presence of the Hlx1 DNA fragment; however, the drastic change of the fluorescence properties of the complex and/or the difficulty of denaturing the complex hindered a similar analysis of the PRDM9–DNA complex.

# Supplementary Information

# Acknowledgements

This work was supported by the "Austrian Science Fund" (FWF) P27698-B22 to I Tiemann-Boege and P31074-B32 to A Horner. Open access funding was provided by Johannes Kepler University Linz. We are grateful to Petko Petkov for providing the vector containing the coding sequence of the *Prdm9* alleles of the strains CAST/EiJ and C57BL/6J, the mouse genomic DNA of these two strains and the vector containing the Halo-ZnF$^{Cst}$ 1–11 construct. We also want to thank Hermann Gruber for fruitful discussions and Angelika Heißl for comments and input in the project.

### Conflict of Interest Statement

The authors declare that they have no conflict of interest.

### Author Contributions

T Schwarz: formal analysis, investigation, methodology, and writing—original draft, review, and editing.
Y Striedner: formal analysis, investigation, methodology, and writing—original draft, review, and editing.

A Horner: formal analysis, investigation, methodology, and writing—review and editing.

K Haase: investigation and methodology.

J Kemptner: formal analysis, investigation, methodology, and writing—original draft.

N Zeppezauer: investigation and methodology.

P Hermann: formal analysis.

I Tiemann-Boege: conceptualization, resources, formal analysis, supervision, funding acquisition, project administration, and writing—original draft, review, and editing.

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
