## [Reviewer comments · Life Science Alliance]

Life Science Alliance

PRDM9 forms a trimer by interactions within the zinc finger array

Theresa Schwarz, Yasmin Striedner, Andreas Horner, Karin Haase, Jasmin Kemptner, Nicole Zeppezauer, Philipp Hermann, and Irene Tiemann-Boege

DOI: [10.26508/lsa.201800291](https://doi.org/10.26508/lsa.201800291)

Corresponding author(s): Irene Tiemann-Boege, Johannes Kepler University, Linz, Austria

Review Timeline:

Submission Date:	2018-12-28
Editorial Decision:	2019-01-03
Appeal Received:	2019-01-07
Editorial Decision:	2019-02-15
Revision Received:	2019-06-10
Editorial Decision:	2019-06-21
Revision Received:	2019-07-02
Accepted:	2019-07-03

Scientific Editor: Andrea Leibfried

Transaction Report:

January 3, 2019

Re: Life Science Alliance manuscript #LSA-2018-00291-T

Irene Tiemann-Boege
1Institute of Biophysics, Johannes Kepler University, Linz, Austria

Dear Dr. Tiemann-Boege,

Thank you for submitting your manuscript entitled "PRDM9 forms an active trimer mediated by its repetitive zinc finger array". The manuscript has been evaluated in depth by me and our editorial decision is not to offer publication of the manuscript in its current form in Life Science Alliance .

I appreciate that PRDM9 plays a key role in regulating and determining the location of meiotic recombination hotspots. It is known to bind DNA at recombination hotspot sequences via the ZnF array domain and to play a role in recruiting the recombination machinery there. PRDM9 dosage is thought to be important for its function. Functional multimerization had been reported in overexpression assays (ref 16, 36). You add to this body of knowledge an well developed set of in vitro binding assays on two different PRDM9 isoforms including domain mapping. You conclude that trimers are formed in vitro on DNA and that a trimer binds a single DNA site (using EMSA and native gel electrophoresis based assays) and also that 5 of the 11 ZnF domains are required for multimerization, and only 1 of the three monomers contact DNA via the ZnF array. Finally, Mass spec confirms no other proteins are involved.

I appreciate the new insight this data provides, but note that it would be more compelling for this journal to show evidence that the trimers bind to single hotspots in vivo. This would broaden the physiological relevance and this the value to the broader meiotic/recombination community. We would not necessarily expect genome wide analysis in vivo.

if you were able to add data along these lines, we would be very pleased to peer review a revised version of the manuscript.

We hope that the comments are useful as your work progresses.

Thank you for thinking of Life Science Alliance as an appropriate place to publish your work.

Sincerely,

Bernd Pulverer

Acting Scientific Editor
Life Science Alliance

February 15, 2019

Re: Life Science Alliance manuscript #LSA-2018-00291-TR-A

Irene Tiemann-Boege
Institute of Biophysics, Johannes Kepler University, Linz, Austria

Dear Dr. Tiemann-Boege,

Thank you for submitting your manuscript entitled "PRDM9 forms an active trimer mediated by its repetitive zinc finger array" to Life Science Alliance. The manuscript was assessed by expert reviewers, whose comments are appended to this letter.

As you will see, the reviewers appreciate your data, but think that additional work is needed to make this manuscript suitable for publication. They provide constructive input on how to do so. We would thus like to invite you to submit a revised version of this work. Importantly, while some of the concerns can get addressed by text changes (see also report of reviewer #2), reviewer #2 points out that the data would also fit with PRDM9 domains binding DNA as a tetramer. Reviewer #1 gives guidance on how to provide better support for the proposed trimeric nature and to distinguish whether multimerization occurs prior or upon DNA binding, so please address these issues. Reviewer #1 also points out that controls are needed to support your conclusions, and these should all get performed. Finally, the experiment aiming at excluding bacterial protein contamination needs to be re-done with conditions closer to your other in vitro conditions (reviewer #1).

Thank you for this interesting contribution to Life Science Alliance. We are looking forward to receiving your revised manuscript.

Sincerely,

B. MANUSCRIPT ORGANIZATION AND FORMATTING:

Reviewer #1 (Comments to the Authors (Required)):

The PRDM9 protein determines meiotic recombination hotspot usage by effecting chromatin modifications around meiotic double strand break sites in most mammals, and also contributes to

speciation via sterility of hybrids with different PRDM9 alleles. The PRDM9 protein contains several domains, in particular a zinc finger array (ZnF array) that mediates protein-DNA sequence recognition, and that is also suggested to mediate protein-protein interaction. Previous work has suggested that PRDM9 acts as a multimer, but direct characterization at the molecular level has not been done.

Schwarz et al. set out to determine the stoichiometry of the DNA-bound PRDM9 complex as well as the number of active DNA binding sites in a complex. In this paper, they use almost exclusively gel-based mobility shift assays (EMSA) that probe the molecular weight of PRDM9-DNA complexes. Using EMSA with DNA containing a single or tandem Hlx1 target sequence (the binding site of the PRDM9) they observed a single shift for a substrate with a single binding site (termed lower shift), and a second shift for the tandem binding site substrate (referred to as a super shift). This in conjunction with another EMSA based experiment using DNA of different lengths is taken as evidence that each PRDM9 complex binds a single target site.

To investigate PRDM9 complex molecular weight (and by extension the number of subunits in the active complex) the authors again used relative migration in EMSA coupled with known standards to determine complex size. The two assays used converge on the active PRDM9-DNA complex containing 3 subunits.

Finally, because their constructs are expressed in bacteria, and many are assayed in unpurified crude bacterial lysates, the authors wanted to exclude the possibility that the binding pattern observed was due to interaction of the DNA with bacterial proteins. Therefore, they performed mass spectrometry on semi-pure DNA-bound ZnF array and determined that no major bacterial proteins were present.

Major points of the paper:

1. Each PRDM9 complex binds a single DNA molecule. This used EMSA assays with DNA targets with either 1 or 2 PRDM9 binding sites. Single-site substrates yielded a single shifted species, which substrates with two tandem binding sites yielded the lower shifted species as well as a predominant second, super-shifted species. Authors estimate that the both binding events have similar KDs, and take this as evidence that for only a single DNA binding site per PRDM9 complex.

The following concerns apply:

a. While this is the most likely explanation, it is possible that the direct tandem binding site arrangement used is incompatible with PRDM9 complex geometry. It would be useful to include at least one substrate where two PRDM9 binding sites are separated by a substantial stretch of DNA.

b. In Figure 1b, there is a small but significant signal above the "supershift" band. Are these the wells, or does this reflect the presence of complexes with more than one DNA molecule?

c. Further evidence for single-DNA binding is given in Figure 4, where the absence of intermediate size with a combination of short and long substrates is taken to argue against two-molecule binding by a single PRDM9 complex. However, it is never shown that intermediate-sized complexes could be resolved on these gels. Inclusion of single-site standards of 150, 348, and possibly 546 bp should be done to establish resolution capabilities. Alternatively, a mixed reaction with a biotinylated 75-mer with an unbiotinylated 273-mer would ask whether any higher MW complexes form with the 75-mer.

2. PRDM9 binds DNA as a trimer. Authors use two EMSA-based assays, both of which infer the size of DNA-protein complexes by comparison to DNA molecules of different sizes. In one assay, the size of the DNA molecules (either one or two sites) is varied; in the other, the size of the protein containing the PRDM9 Zn fingers is varied. Both assays converge, with considerable variance (from

~2.3 to 4), on 3 copies of PRDM9 per bound complex. In addition, these experiments show that ZnFs 2-6 alone are sufficient to bind a DNA target and to form a trimer.

a. This approach is justified by an extensive theoretical discussion of reputation and migration through gels. It is highly unlikely that one or two protein trimers bound to a single DNA molecule approximate a single polymer chain under native conditions. Therefore, this theoretical discussion should be considerably shortened, if not eliminated. The fact remains that, under the gel conditions used here, it appears that protein/DNA complex migration is a good approximate measure for total complex size, and this empirical observation should be sufficient.

b. An important limitation to both of the assays used to determine complex stoichiometry is that the authors can only measure subunit composition for the DNA-bound PRDM9 complex, and thus are unable to address whether PRDM9 binds DNA as a trimer or if it multimerizes upon DNA binding. In Supplementary Figure 4, the authors were unable to detect unbound protein using Coomassie staining, although the DNA-protein complex migrated as a single species. This suggests that, in the absence of DNA, at least this particular PRDM9 construct has a disperse stoichiometry, which might be confirmed using more sensitive protein detection methods, such as silver staining or reagents that react with the Halo tag.

c. As further confirmation that PRDM9 binds DNA as a trimer, authors could either mix extracts or co-express long and short PRDM9 variants; the prediction would be that a single-site DNA substrate would then reveal complexes with four different mobilities (short-short-short, long-short-short, long-long-short, long-long-long), as long as PRDM9 monomers are freely exchanging.

3. Authors perform mass spectrometry on protein-DNA complexes to exclude participation of bacterial proteins, but the protein used was extensively purified before analysis, which compromises the generality of this conclusion, especially since most assays are performed with crude bacterial lysates. Since the DNA substrates are biotinylated, an alternative approach would use that moiety as an affinity tag to pull out from crude lysates and analyze PRDM9-DNA complexes.

Minor points:

4. The paper title is "PRDM9 forms an active timer mediated by its repetitive zinc finger array," but it is not clear what they "active" means; this word could be removed from the title without loss of meaning.

5. In the discussion section "Multimerization of PRDM9 is not exclusive of heteromers" the authors explain co-immunoprecipitation experiments previously performed to analyze homo- and heteromeric complexes in great detail. I suggest this section be trimmed down significantly to leave out the specific details of past tags used, and just state the main conclusion of that study in a few sentences.

6. In the discussion section "The PRDM9 trimer binds only one DNA target" the authors discuss implications of trimerization and binding a single site. This included a discussion regarding chromatin modifications flanking PRDM9 sites. This portion of the discussion is not supported by any of their experimental data and could be shortened and/or removed.

7. The EMSA gels in Figure 1 should be clearly labeled to indicate the wells and include size standards that correspond to: free DNA, PRDM9-bound DNA, double PRDM9-bound DNA, PRDM9 bound to two DNA molecules with tandem sites, and two DNA molecules bound to two PRDM9 sites. The binding curves should also include ones in which (1- fraction unbound) is plotted on the y-axis to determine whether similar binding constants are observed when analyzing the gels in a slightly different manner. Also, rather than using triangles, why not label each lane with the actual protein concentration?

8. The concentration of DNA substrate included in reactions should be clearly stated in either the methods or figure legends. Since protein is in considerable excess over DNA substrate when total concentrations approach the K_d in the experiments here, it is perhaps not surprising that a 1:1

complex:DNA binding ratio is observed. It would be interesting to increase DNA concentrations so that DNA, rather than protein complex, is in excess.

Reviewer #2 (Comments to the Authors (Required)):

The paper by Schwrtz et al. describes a series of invitro experiments showing that the isolated zinc finger domain of the recombination protein PRDM9 forms trimers. This is a matter of some importance e as PRDM9 is an essential protein in meiosis and multimerization has a significant impact on how PRDM9 functions to determine the location of meiosis recombination hotspots when animals or humans are heterozygous, carrying two different alleles of PRDM9.

My general impression of this paper is positive. It deserves to be published, but would certainly benefit from addressing the issues I mention below. It would also be helpful if it could be translated from "German English" into "American/British English".

I have several major caveats, which need to be corrected. The first is that throughout the paper the authors conflate the behavior and properties of the ZNF domain with the properties of the intact protein. The implicit assumption when conflating the two is that other domains of PRDM9, for example the KRAB domain, do not participate in multimer formation, which may very well not be true.

My second concern is the apparent contradiction between the results of references 36 and 71 described in lines 305-325. These references report that PRDM9 does and does not form multimers in vivo. What is the authors interpretation of these data, a matter that seems to be central to their thesis.

And third, although the authors claim the formation of trimers, the data seem equally compatible with tetramers. Unless there is evidence to the contrary, it would seem best to equivocate on the exact stoichiometry of the complexes.

73 - not quite true. Baker et al. Ref 36 showed it to be the case in vivo in mouse testis.

98 - "then there would be no change" is more appropriate phrasing.

100 - same here. "would have" is the right phrasing.

102 - It would be better to write "Thus we used the DNA sequence containing...".

107-123 - It would be appropriate if they added an experiment which the control had the same size as the single site oligo. Figure 4 seems to partially address this objection but it is not well utilized in the paper.

173-182 - As justification for using selected domains for their experiments the authors might want to cite the insolubility of intact PRDM9.

181 - "Partially purified" would be the more appropriate term

183 - "whether" would be more appropriate than "that"

209- "non-specific" would be more appropriate

220 - delete "mainly"

218-241 - these data provide good evidence that it is only PRDM9-DNA interactions that play a role in the in vitro experiments but they are described in such an incomprehensible way that it is hard work to evaluate them. It would be best if this could be rewritten to make it easier to understand. What I see as a potential problem is that only the peptides "HQRTHTGEKPYVCR" and "SFIASEISSIER" are part of the ZnF construct; the other two are tag-based sequences.

232-233 - what is non-specific

305-325 - these paragraphs are confusing in several ways. They mix data from in vitro experiments using isolated zinc finger domains with cellular and organismal data describing the behavior of intact PRDM9. They do not make it clear that multiimerization of the zinc finger domain in vitro must involve the zinc fingers, while multimerization in cells can involve other domains of PRDM9,

particularly the KRAB domain, which is known to be active in protein: protein interactions. The second confusion relates to the fact that the experiments of ref 36 (lines 314-316) and ref 71 (lines 321-323) contradict each other. No mention of the existence or possible explanations of this discrepancy are presented, although it would seem to be central to the question of whether PRDM9 forms multimers.

Figure 2. The entire figure shows that PRDM9-DNA ratio is about 3.5. Therefore, a more cautious evaluation throughout the paper stating that PRDM9 binds DNA as either trimer or tetramer would be more appropriate. In fact, given that PRDM9's pI (for all constructs, including full-length and ZnF only) is between 8.9 and 9.1, a more conservative assessment and mentioning these facts would be good. The authors discuss this earlier stating that the longer DNA oligo size determines the EMSA mobility according to MW, but I am not convinced by the argumentation.

Response to reviewers

We would first like to thank all reviewers for their helpful and constructive comments that have substantially improved the MS. We revised the manuscript integrating the reviews and recommendations of the reviewers (in black) and have addressed each of the reviewer's points below (in green). In brief, the biggest changes, which also include new experiments, can be summarized in the following:

1. We added an experiment performed with fluorescence correlation spectroscopy (FCS) to further explore the trimeric nature of PRDM9. In particular, we used FCS to estimate the fluorescent particle number of soluble, free eYFP-PRDM9 before and after denaturation with urea (see Figure 3C, Figure S6, Table 2, Table S5, Materials and Methods, Supplementary Materials). We estimated with FCS that the PRDM9 is formed by three particles supporting the model that PRDM9 is a trimer that does not change its stoichiometry upon DNA binding.
2. We present in Figure S2 an additional experiment showing that two complexes bind on the tandem DNA fragment carrying two consecutive binding sites. In brief, by adding a restriction enzyme site on this DNA fragment, we showed that the super-shift can be separated by a restriction enzyme digest into two independent protein-DNA complexes.
3. We extended the number of data points presented in Figure 4 with three additional combinations of long and short DNA fragments to fully validate our observation that the PRDM9 trimer just binds one DNA molecule at a time (Figure S9).
4. We added an experiment (see Figure S7), in which we co-expressed two different PRDM9-ZnF constructs of different MWs. Our results indicate that there is no exchange between the monomers from the different constructs suggesting that multimerization occurs already before binding to the DNA.
5. We rearranged the presentation of the multimerization analysis by EMSA in terms of experiments (Figure 3, Figure S5), construct types, and expression systems. This should help the reader to better understand and interpret the results. Moreover, we mainly base our conclusions on experiments of the more accurate measurements from *assay II*.

Changes made in the manuscript are written in green text in the revised files.

The individual concerns raised by the reviewers are dissected point by point in the next paragraphs.

Reviewer #1 (Comments to the Authors (Required)):

We thank the reviewer for the time in carefully reviewing our manuscript.

The PRDM9 protein determines meiotic recombination hotspot usage by effecting chromatin modifications around meiotic double strand break sites in most mammals, and also contributes to speciation via sterility of hybrids with different PRDM9 alleles. The PRDM9 protein contains several domains, in particular a zinc finger array (ZnF array) that mediates protein-DNA sequence recognition, and that is also suggested to mediate protein-protein interaction. Previous work has suggested that PRDM9 acts as a multimer, but direct characterization at the molecular level has not been done.

Schwarz et al. set out to determine the stoichiometry of the DNA-bound PRDM9 complex as well as the number of active DNA binding sites in a complex. In this paper, they use almost exclusively gel-based mobility shift assays (EMSA) that probe the molecular weight of PRDM9-DNA complexes. Using EMSA with DNA containing a single or tandem Hlx1 target sequence (the binding site of the PRDM9) they observed a single shift for a substrate with a single binding site (termed lower shift), and a second shift for the tandem binding site substrate (referred to as a super shift). This in conjunction with another EMSA based experiment using DNA of different lengths is taken as evidence that each PRDM9 complex binds a single target site.

To investigate PRDM9 complex molecular weight (and by extension the number of subunits in the active complex) the authors again used relative migration in EMSA coupled with known standards to determine complex size. The two assays used converge on the active PRDM9-DNA complex containing 3 subunits.

Finally, because their constructs are expressed in bacteria, and many are assayed in unpurified crude bacterial lysates, the authors wanted to exclude the possibility that the binding pattern observed was due to interaction of the DNA with bacterial proteins. Therefore, they performed mass spectrometry on semi-pure DNA-bound ZnF array and determined that no major bacterial proteins were present.

Major points of the paper:

1. Each PRDM9 complex binds a single DNA molecule. This used EMSA assays with DNA targets with either 1 or 2 PRDM9 binding sites. Single-site substrates yielded a single shifted species, which substrates with two tandem binding sites yielded the lower shifted species as well as a predominant second, super-shifted species. Authors estimate that the both binding events have similar K_D s, and take this as evidence that for only a single DNA binding site per PRDM9 complex. The following concerns apply:

1.a. While this is the most likely explanation, it is possible that the direct tandem binding site arrangement used is incompatible with PRDM9 complex geometry. It would be useful to include at least one substrate where two PRDM9 binding sites are separated by a substantial stretch of DNA.

We wanted to show with the tandem fragment experiments that two independent PRDM9 complexes are bound on the fragment (observed as a super-shift), each occupying one of the two adjacent binding sites. Given the large difference in migration distance between the lower-shift and the super-shift and the similar K_D between these two shifts, it is likely that this is the case; however, we agree with the reviewer that we should provide further proof.

For this purpose, we did an additional experiment in which we separated the two binding sites by a restriction enzyme site (whether this spacer is enough to allow for a more relaxed complex geometry is not known). As expected, this fragment also migrated as a lower-shift and a super-shift. Note also a third, fast migrating band that we cannot explain, but could be due to a tighter folding. However, this is not the first time we obtained an unexplainable banding pattern when rearranging the binding site within the DNA. However, when digesting the tandem fragment after PRDM9 binding with a restriction enzyme, the super-shift was gone and we observed instead two lower-shifts now with the same migration distance as the two short fragments with one binding site. This proves that the super-shift carries two independent complexes that can be separated by a restriction enzyme digest. We describe this experiment in L102-116 and added the data to Supplementary_Materials, Figure S2.

However, we think that the point that the reviewer wants to make here is about the interaction of the PRDM9-ZnF oligomer with multiple DNA binding sites. The reviewer is concerned that the tandem

data does not prove that multiple ZnF arrays within a multimer do not interact simultaneously with several DNA binding sites, since in the originally tandem this might be “incompatible with PRDM9 complex geometry”. We agree and for this reason we already had performed the experiment presented in Figure 4, in which we added different sizes of DNA (long and short) simultaneously to our binding reaction (see also response to point 1c). Please note that we extended this experiment to several additional fragment combinations now, and unequivocally observed for all combinations that each DNA-protein complex forms with one or the other DNA, but not both (Figure S9). We now better refer to this experiment (Figure 4) within this first section in L115-116. See also comments to point 1c. We also added binding data with increasing DNA concentrations (representing saturation conditions)—see response to point 8.

1.b. In Figure 1b, there is a small but significant signal above the "supershift" band. Are these the wells, or does this reflect the presence of complexes with more than one DNA molecule?

These are the wells of the EMSA gel. We sometimes observed these bands in all sorts of different experiments, especially when working with very “dirty” extracts. We believe these represent our labelled DNA bound to big unspecific agglomerates (also with other unspecific DNA binding proteins) that stick in the slots and cannot run into the gel properly. We also observed that polydIdC reduces somewhat this signal. Note that this band is absent in other experiments with other tandem fragments (e.g. Figure S3).

1.c. Further evidence for single-DNA binding is given in Figure 4, where the absence of intermediate size with a combination of short and long substrates is taken to argue against two-molecule binding by a single PRDM9 complex. However, it is never shown that intermediate-sized complexes could be resolved on these gels. Inclusion of single-site standards of 150, 348, and possibly 546 bp should be done to establish resolution capabilities.

It is rather unlikely that with our set-up we cannot resolve intermediate fragments. In fact, we obtained an almost perfect correlation (99%) between migration distance and MW of complexes on DNA ranging from 75bp to 1460bp of 10 different complexed fragments separated by ~100bp-steps (see Figure 2C). This excellent correlation indicates that we can resolve a complex with a 75bp fragment from a complex with a 114bp fragment, or a 150bp fragment, and so on. A complex with a 352bp fragment can also be distinguished from a 468bp-complex (both with a MW difference that is smaller than expected between the intermediate fragments of 348 and 546bp).

However, we agree that one fragment combination might not be fully convincing to exclude a two-molecule-binding model for whatever other unknown reasons. Thus, we performed three more fragment combinations (shown in Figure S9) with shorter and longer DNA fragments (75 + 189, 189 + 856, and 273 + 543). In none of these combinations, we see intermediate-sized bands confirming that the PRDM9 trimer binds only one molecule at a time, as proposed in model 1. We describe the results of this experiment in L273-278.

Alternatively, a mixed reaction with a biotinylated 75-mer with an unbiotinylated 273-mer would ask whether any higher MW complexes form with the 75-mer.

The suggestion of mixing unlabeled and labelled DNA of different sizes is an interesting idea. This experiment is already published, albeit with a different purpose (see Striedner et al. 2017 Chromosome Research, Figure 2). In that experiment we added a hot 75bp (labelled) fragment to an 100-fold excess cold 39bp (unlabeled) competitor or Figure 3 in which PRDM9 was incubated with several different

short, cold fragments added at increasing concentrations to a constant amount of long, hot DNA. We did not see a difference in the migration distance of the complex with the hot 75bp fragment with or without the cold competitor (but as expected the intensity of the shift changed). Therefore, we can rule out that intermediate complexes form in the presence of these two populations of DNA. We now included an un-published EMSA of one of these experiments in Figure S10. See also response to comment 8.

2. PRDM9 binds DNA as a trimer. Authors use two EMSA-based assays, both of which infer the size of DNA-protein complexes by comparison to DNA molecules of different sizes. In one assay, the size of the DNA molecules (either one or two sites) is varied; in the other, the size of the protein containing the PRDM9 Zn fingers is varied. Both assays converge, with considerable variance (from ~2.3 to 4), on 3 copies of PRDM9 per bound complex. In addition, these experiments show that ZnFs 2-6 alone are sufficient to bind a DNA target and to form a trimer.

We are aware that some of the measurements underlie a large variance. However, this is a problem mainly in *assay I*, which is less accurate than *assay II*. There is also a plausible explanation for this: the migration distance changes exponentially with MW, resulting in less accurate measurements and a larger variance for the higher MW estimates of the tandem fragments used for *assay I* (see Table 1, Figure S3 and Figure S5)). For this reason, we decided to rely mainly on the measurements of *assay II* that clearly show that the MW for the complex is of three units, except for two out of eight constructs (e.g. eYFP-ZnF 1-11 and ZnF 2-8). This point is revised in L1171-176 and L181-185.

2.a. This approach is justified by an extensive theoretical discussion of reptation and migration through gels. It is highly unlikely that one or two protein trimers bound to a single DNA molecule approximate a single polymer chain under native conditions. Therefore, this theoretical discussion should be considerably shortened, if not eliminated. The fact remains that, under the gel conditions used here, it appears that protein/DNA complex migration is a good approximate measure for total complex size, and this empirical observation should be sufficient.

We appreciate the reviewer's view that our data itself are convincing enough to show that we can infer the MW of a protein complex based on migration distances under non-denaturing conditions. However, in light of the dogma that the migration distance is proportional to the MW in denatured/linear polymer chains, we felt compelled to explain further the validity of our observations. Thus, we would like to leave the theoretical "reptation" model in the manuscript.

2.b. An important limitation to both of the assays used to determine complex stoichiometry is that the authors can only measure subunit composition for the DNA-bound PRDM9 complex, and thus are unable to address whether PRDM9 binds DNA as a trimer or if it multimerizes upon DNA binding.

We really appreciate this comment and we have been very keen into exploring this point further, but did not have previously the means to do so. We are very excited to have now further data on this aspect. We used fluorescence correlation spectroscopy (FCS) to estimate the number of fluorescent units in our recombinant eYFP-labelled PRDM9 construct. In order to infer the number of fluorescent units with FCS, we compared the FCS autocorrelation of the labelled protein in native (1xTKZN buffer) and denaturing conditions (3M urea). We observed ~3 times as many particles in the focal volume in the denatured form (monomer) than in the native eYFP-PRDM9 supporting the trimeric nature of the native oligomer (see Figure 3 and Figure S6). In comparison, a similar construct only with eYFP (mainly a monomer in its native state) showed hardly a change in particle number upon

denaturation. The FCS data are congruent with our EMSA measurements on the trimeric nature of the complex. Moreover, it seems that PRDM9 does not change its stoichiometry upon DNA binding. We added this experiment in the MS in L198-215.

2.c. In Supplementary Figure 4, the authors were unable to detect unbound protein using Coomassie staining, although the DNA-protein complex migrated as a single species. This suggests that, in the absence of DNA, at least this particular PRDM9 construct has a disperse stoichiometry, which might be confirmed using more sensitive protein detection methods, such as silver staining or reagents that react with the Halo tag.

We disagree with the interpretation of the reviewer (disperse stoichiometry), since native gel electrophoresis cannot be used to compare the migration of bound or unbound PRDM9 and infer a stoichiometry. Note that if a native protein is in a pH region below its isoelectric point (pI), then it will be positively charged and will migrate towards the negative pole in electrophoresis. In our case, the pH of the electrophoresis buffer was 7.5 and the pI of our proteins was >8.0 . All of our native constructs are positively charged and migrate towards the negative pole. Thus, we do not expect to observe a band for the unbound PRDM9 in the gel. We controlled in this Coomassie gel that indeed this is the case.

However, when our protein constructs are bound to DNA (PRDM9+Hlx1), then the overall charge of the complex is defined by the excess negative charges of the DNA driving the migration of the complex (PRDM9+DNA) into the gel towards the positive pole.

Also note, that the purpose of this experiment was to cut out the band of the complex and we used the Coomassie staining to orient ourselves as to the position of this band and to roughly estimate the amount of the complex added to the mass spectrometer.

2.d. As further confirmation that PRDM9 binds DNA as a trimer, authors could either mix extracts or co-express long and short PRDM9 variants; the prediction would be that a single-site DNA substrate would then reveal complexes with four different mobilities (short-short-short, long-short-short, long-long-short, long-long-long), as long as PRDM9 monomers are freely exchanging.

This is also an interesting suggestion, which we attempted (Figure S7). In short, we co-expressed two different variants of ZnF^{Cst} with the same number of zinc fingers, but different MW given by an eYFP tag (eYFP-ZnF^{Cst} 2-11 and ZnF^{Cst} 2-11 with 62 and 37 kDa, respectively). These two constructs, carried by two different plasmid vectors, were co-transfected in equimolar amounts. Both constructs were expressed at similar concentrations verified by a Western blot. However, in EMSA we did not observe a mixture of short and long intermediate shifts. Instead, we observed with this co-expressed PRDM9-ZnF mixture only two shifts: each at the same migration distance expected for one of the PRDM9-ZnF constructs (ran in parallel). This suggests that PRDM9 forms already a stable multimer before DNA binding, congruent also with our FCS observations (see response 2b). We describe this experiment in L216-226. Unfortunately, we cannot use this experiment to further validate the trimeric nature of PRDM9 (as suggested by the reviewer), given that this set-up does not fulfill the premise that “PRDM9 monomers are freely interchanging”.

The results of this experiment were quite surprising since the Co-IP studies of Altemose et al 2017 and Baker et al 2015 proof the formation of mixtures of co-expressed PRDM9 carrying different tags or PRDM9 types. It could be possible that our system is not sensitive enough or did not provide the required conditions to allow the observation of mixtures of co-expressed constructs. Moreover, it was unclear if previous studies co-transfected independent plasmids or used one plasmid with several open reading frames, which could play a role in mixing different monomers within one cell.

3. Authors perform mass spectrometry on protein-DNA complexes to exclude participation of bacterial proteins, but the protein used was extensively purified before analysis, which compromises the generality of this conclusion, especially since most assays are performed with crude bacterial lysates. Since the DNA substrates are biotinylated, an alternative approach would use that moiety as an affinity tag to pull out from crude lysates and analyze PRDM9-DNA complexes.

We disagree with the claim that the mass spec analysis is limited to only one particular construct. Our stoichiometry inferences are based on the migration distance of a shift representing the complex PRDM9-DNA. We showed with many different experiments that the only parameter affecting the mobility in our system was the MW of the complex. Thus, we can make the generalized claim that none of the different parameters such as expression types, tags, purification or additives (e.g. polydIdC) affected the mobility in EMSA and a complex cut out from a gel for mass spec analysis should represent any of the other shifts, regardless of how the construct was initially produced.

Moreover, gel electrophoresis could be considered as a purification step already by itself, since only proteins bound to our biotinylated DNA migrated into the gel and were analyzed (our control without DNA did not show a band at this height—see also point 2c). Thus, crude lysates, soluble fractions in the supernatant, or *in vitro* cell expression systems, etc. could be considered “purified” forms of protein when complexed to the DNA in the shift.

However, we appreciate the suggested experiment by the reviewer. In fact, we initially attempted to capture protein-DNA complexes via biotinylated DNA attached to magnetic streptavidin beads that was released again by a DNA digest. Unfortunately, with this approach we did not obtain enough material for mass spectrometry (data not shown). Thus, we extracted the complex from our native gel-electrophoresis using the Halo tagged construct. This system had several advantages: first, we obtained much higher yields than with the biotin capture method improving considerably the mass spectrometry data. Second, we did not require the addition of polydIdC to block the binding of unspecific proteins to our DNA, as done for larger volumes of “crude” extracts. Third, this procedure (cutting out the shift) represents the analysis of proteins within the shift.

We rephrased the text now to convey that we used mass spectrometry to assess the protein composition of the shift, representative of any shift in our system regardless of the initial construct conditions (see L228-237).

Minor points:

4. The paper title is "PRDM9 forms an active trimer mediated by its repetitive zinc finger array," but it is not clear what they "active" means; this word could be removed from the title without loss of meaning.

We modified the title now to “PRDM9 forms a trimer by interactions within the zinc finger array.”

5. In the discussion section "Multimerization of PRDM9 is not exclusive of heteromers" the authors explain co-immunoprecipitation experiments previously performed to analyze homo- and heteromeric complexes in great detail. I suggest this section be trimmed down significantly to leave out the specific details of past tags used, and just state the main conclusion of that study in a few sentences.

We re-organized this whole section and included also a simpler description of the multimerization experiments in the different publications.

6. In the discussion section "The PRDM9 trimer binds only one DNA target" the authors discuss implications of trimerization and binding a single site. This included a discussion regarding chromatin

modifications flanking PRDM9 sites. This portion of the discussion is not supported by any of their experimental data and could be shortened and/or removed.

What we meant to say was that the fact that the ZnF array binds only to one DNA target does not preclude that other domains of the PRDM9 multimer are not active. We re-phrased this paragraph to better convey this point and removed the discussion about chromatin modifications flanking PRDM9.

7. The EMSA gels in Figure 1 should be clearly labeled to indicate the wells and include size standards that correspond to: free DNA, PRDM9-bound DNA, double PRDM9-bound DNA, PRDM9 bound to two DNA molecules with tandem sites, and two DNA molecules bound to two PRDM9 sites. The binding curves should also include ones in which (1- fraction unbound) is plotted on the y-axis to determine whether similar binding constants are observed when analyzing the gels in a slightly different manner. Also, rather than using triangles, why not label each lane with the actual protein concentration?

We changed Figure 1 using more intuitive labels such as ‘free DNA (single-Hlx1)’ and ‘single-complex’ (unbound and bound DNA in lower-shift, respectively) for Figure 1A and ‘free DNA (tandem-Hlx1)’, ‘single-complex’ and ‘double-complex’ (unbound and bound DNA in super- or lower-shift, respectively) for Figure 1B. Note that the EMSA shown in Figure 1 is a subset of the full data presented now in Table S1 listing all protein concentrations (the range is now given in the legend in order to avoid overcrowding of the figure). The binding affinity needs to be estimated as we have done, and this approach was validated already in Striedner et al 2017. However, in panel C and D of Figure 1 we do a similar analysis as suggested and show the curves of fractions DNA free (grey) DNA bound by 1 complex (red) and DNA bound by 2 complexes (purple; exclusive for using tandem-Hlx1 DNA in panel D).

8. The concentration of DNA substrate included in reactions should be clearly stated in either the methods or figure legends. Since protein is in considerable excess over DNA substrate when total concentrations approach the K_d in the experiments here, it is perhaps not surprising that a 1:1 complex:DNA binding ratio is observed. It would be interesting to increase DNA concentrations so that DNA, rather than protein complex, is in excess.

We made sure that all the important concentrations of DNA and protein used in individual experiments are stated in the figure legends. All additional details about reaction substrates and conditions are shown in supplementary tables and methods.

We now also included a figure that shows that PRDM9 binds only to one molecule even at conditions using an excess of DNA (see Figure S10). This EMSA is unpublished, albeit the data was used for a different purpose in Striedner et al 2017 and Tiemann-Boege et al 2017. In short, we incubated PRDM9 with a constant amount of hot, long DNA (75bp) and an increasing amount of cold, short DNA (39bp) reaching an excess of DNA vs protein. The mobility of the complex does not change with increasing DNA concentrations, which would be expected if more than one DNA (75bp hot DNA long and 39bp cold DNA short) would be bound. Note that in this experiment we do not observe two shifts, since the shorter DNA (titrated from 0 to 1500nM) was not biotinylated (cold) to avoid smearing in the EMSA.

Reviewer #2 (Comments to the Authors (Required)):

The paper by Schwrtz et al. describes a series of invitro experiments showing that the isolated zinc finger domain of the recombination protein PRDM9 forms trimers. This is a matter of some importance as PRDM9 is an essential protein in meiosis and multimerization has a significant impact on how PRDM9 functions to determine the location of meiosis recombination hotspots when animals or humans are heterozygous, carrying two different alleles of PRDM9. My general impression of this paper is positive. It deserves to be published, but would certainly benefit from addressing the issues I mention below. It would also be helpful if it could be translated from "German English" into "American/British English".

We apologize that in the Table S6 we used German terminology for some of the chemical compounds, which was the default software language of the mass spec analysis. We have corrected this now.

I have several major caveats, which need to be corrected.

1. The first is that throughout the paper the authors conflate the behavior and properties of the ZNF domain with the properties of the intact protein. The implicit assumption when conflating the two is that other domains of PRDM9, for example the KRAB domain, do not participate in multimer formation, which may very well not be true.

In this work, we mainly focused on the ZnF array, but included in our analysis also the three other domains in one of our constructs (full-length construct-see YFP-PRDM9^{Cst}; 127kDa). Of course, with this experiment we cannot exclude if the other domains (KRAB, SSXR, PR/SET individually or together) play an additional role in the multimerization. However, we clearly show that multimer formation is primarily mediated by the ZnF array. By comparing the full length with a series of shortened constructs of a few ZnFs, we show that the KRAB, SSXR, PR/SET domains are NOT necessary for the multimer formation and that five ZnFs within the DNA binding domain are already sufficient to induce multimerization. We rephrased this point better in the manuscript (L181-189). We also added the observation of Altemose et al 2017 that a weak multimerization can be observed without the ZnF domain.

2. My second concern is the apparent contradiction between the results of references 36 and 71 described in lines 305-325. These references report that PRDM9 does and does not form multimers in vivo. What is the authors interpretation of these data, a matter that seems to be central to their thesis. As to the interpretation of the contradiction of different works, it is not in the scope of this work to address this, but should be done in a review. As to our personal opinion, it is difficult to assess fully the weight of the contradiction about multimerization of one work (ref 71) versus two others (three including this one). We interpret the "lack of multimerization" in ref 71 as lack of heteromers (related maybe to the preferential formation of a homomer, or/and the lack of exchange between monomers once formed), which, does not exclude the possibility of multimerization *per se*. We added a short comment in this regard to the MS.

3. And third, although the authors claim the formation of trimers, the data seem equally compatible with tetramers. Unless there is evidence to the contrary, it would seem best to equivocate on the exact stoichiometry of the complexes.

Several of our data sets suggest that the stoichiometry is a trimer (EMSA and FCS), but not a tetramer or a dimer. We agree that some of the data has large variations (e.g. EMSA *assay I*). However, the large MW of the constructs used in this assay can account for this. The accuracy of inferring the MW from migration distances decreases rapidly with molecules of large MW, given the inverse exponential correlation of migration distance with MW in gel electrophoresis. Thus, we base our conclusions on the more accurate EMSA *assay II*. In addition, we have now FCS data, which is based on a completely different set-up, and also corroborates the trimeric nature of PRDM9. We now include these aspects in the MS (also see response to reviewer 1, point 2).

4. 73 - not quite true. Baker et al. Ref 36 showed it to be the case *in vivo* in mouse testis.

We closely examined the Baker et al paper (ref 36) and to our knowledge/interpretation, the evidence for PRDM9 multimerization is derived from experiments co-expressing different PRDM9 alleles in HEK293 cells. The authors used murine testicular samples for indirect immunofluorescence labeling to test for *Prdm9* dosage sensitivity and its impact on the male meiotic progression *in vivo* of heterozygous null mice. However, we could not find any data about multimerization related to murine testis samples.

5. 98 - "then there would be no change" is more appropriate phrasing.

We changed the phrasing accordingly.

6. 100 - same here. "would have" is the right phrasing.

We changed the phrasing accordingly.

7. 102 - It would be better to write "Thus we used the DNA sequence containing...".

We changed the phrasing accordingly.

8. 107-123 - It would be appropriate if they added an experiment which the control had the same size as the single site oligo. Figure 4 seems to partially address this objection but it is not well utilized in the paper.

We now added an experiment showing that the super-shift is indeed formed by two complexes, each bound to one of the two binding sites. In short, we separated the two binding sites by a restriction enzyme site. When digesting this new tandem fragment after PRDM9 binding with a restriction enzyme, the super-shift was gone and we observed instead two lower-shifts, at the same migration distance as for the two short fragments with one binding site. This proves that the super-shift carries two independent complexes that can be separated by a restriction enzyme digest. We describe this experiment in L102-116 and added the data to Figure S2.

We also increased the number of experiments shown in Figure 4. These additional experiments (Figure S9) all corroborate that the PRDM9 trimer binds just one DNA molecule (see also response to reviewer 1; 1a and 1c).

9. 173-182 - As justification for using selected domains for their experiments the authors might want to cite the insolubility of intact PRDM9.

We chose all these different constructs to show that different MWs, pIs and expression conditions do not influence the nature of the multimerization. Insolubility was not an issue in this case; although, we agree that some constructs were more soluble than others including ones not used here, but this also depends on many other factors other than the domains of the protein.

10. 181 - "Partially purified" would be the more appropriate term

We changed the wording accordingly.

11. 183 - "whether" would be more appropriate than "that"

We changed the wording accordingly.

12. 209- "non-specific" would be more appropriate

We changed the wording accordingly.

13. 220 - delete "mainly"

We deleted 'mainly'

14. 218-241 - these data provide good evidence that it is only PRDM9-DNA interactions that play a role in the in vitro experiments but they are described in such an incomprehensible way that it is hard work to evaluate them. It would be best if this could be rewritten to make it easier to understand. What I see as a potential problem is that only the peptides "HQRHTHTGEKPYVCR" and "SFIASEISSIER" are part of the ZnF construct; the other two are tag-based sequences.

We apologize for the incomprehensible wording and rephrased this section for more clarity. The main point of the mass-spec analysis was to assure that no other large peptides (e.g. derived from bacteria) were part of the complex and would potentially introduce an error in the inferred stoichiometry. That two of these peptides are not part of the ZnF domain, but are instead of the tag is in agreement that these belong to the amino acid sequence of the used construct. Also note that many more peptides were analyzed in Table S6 and covered ~60% of the amino acid sequence of our protein construct.

15. 232-233 - what is non-specific

We rephrased this accordingly

16. 305-325 - these paragraphs are confusing in several ways. They mix data from in vitro experiments using isolated zinc finger domains with cellular and organismal data describing the behavior of intact PRDM9. They do not make it clear that multimerization of the zinc finger domain in vitro must involve the zinc fingers, while multimerization in cells can involve other domains of PRDM9,

particularly the KRAB domain, which is known to be active in protein: protein interactions. The second confusion relates to the fact that the experiments of ref 36 (lines 314-316) and ref 71 (lines 321-323) contradict each other. No mention of the existence or possible explanations of this discrepancy are presented, although it would seem to be central to the question of whether PRDM9 forms multimers.

We reorganized this whole section and re-wrote several parts, also highlighting the role of the other domains.

17. Figure 2. The entire figure shows that PRDM9-DNA ratio is about 3.5. Therefore, a more cautious evaluation throughout the paper stating that PRDM9 binds DNA as either trimer or tetramer would be more appropriate. In fact, given that PRDM9's pI (for all constructs, including full-length and ZnF only) is between 8.9 and 9.1, a more conservative assessment and mentioning these facts would be good. The authors discuss this earlier stating that the longer DNA oligo size determines the EMSA mobility according to MW, but I am not convinced by the argumentation.

We rearranged the presentation of the stoichiometry better represented in Figure 3, which includes now only the more accurate data of assay II, which unequivocally shows a stoichiometry of three units. The full data set is now presented in Table S2 and S3. We also reevaluated our conclusions in light of new FCS data that is also congruent with a trimer structure (see also response to reviewer 1; point 2 and 2b). The purpose of Figure 2 was to show the principle of the assay, but not to convey a stoichiometry. See also response to comment reviewer 1 (2 and 2b).

June 21, 2019

RE: Life Science Alliance Manuscript #LSA-2018-00291-TRR

Dr. Irene Tiemann-Boege
Johannes Kepler University, Linz, Austria
Institute of Biophysics
Gruberstrasse 40
Linz 4020
Austria

Dear Dr. Tiemann-Boege,

Thank you for submitting your revised manuscript entitled "PRDM9 forms a trimer by interactions within the zinc finger array". As you will see, the reviewers appreciate the introduced changes. We would be thus happy to publish your paper in Life Science Alliance pending final revisions necessary to address a few remaining concerns of the reviewers and formatting issues:

- please address the remaining reviewer concerns by text changes
- please add a callout in the text to Fig S3C
- please mention in the legend to figure S3A that the same blot is shown as the one in Fig2B
- please note that I have separated the S figures from your supplementary information file as these will be displayed in-line in the HTML version of your paper

A. FINAL FILES:

- An editable version of the final text (.DOC or .DOCX) is needed for copyediting (no PDFs).
- High-resolution figure, supplementary figure and video files uploaded as individual files: See our detailed guidelines for preparing your production-ready images, <http://www.life-science-alliance.org/authors>
- Summary blurb (enter in submission system): A short text summarizing in a single sentence the

study (max. 200 characters including spaces). This text is used in conjunction with the titles of papers, hence should be informative and complementary to the title. It should describe the context and significance of the findings for a general readership; it should be written in the present tense and refer to the work in the third person. Author names should not be mentioned.

B. MANUSCRIPT ORGANIZATION AND FORMATTING:

Sincerely,

Reviewer #1 (Comments to the Authors (Required)):

The authors have done a good job of addressing our concerns, including the addition of FCS experiments which convincingly reinforce the conclusion that their expressed protein is a trimer in solution. Our one remaining concern is with regard to the failure to detect mixture of long and short isoforms amongst co-expressed proteins, contrary to observations of others. While this should not interfere with proceeding to publication, authors are urged to explore coexpression from a single plasmid or urea denaturation/reannealing as a way of constructing mixed oligomers.

Minor comment--to avoid confusion, location of gel wells should be indicated on EMSA gels showing signal at the well positions (Figure 1b).

Reviewer #3 (Comments to the Authors (Required)):

The authors have addressed all issues identified by the reviewers and added new experimental evidence supporting their main conclusions. I now find the manuscript to be suitable for publication in Life Science Alliance. I would still advise them to be more cautious and balanced when addressing the nature of interactions between PRDM9 monomers. Although the authors provide compelling evidence that such interactions can be mediated by identical ZNF domains, this is not the only possibility. Patel et al (2016, Genes Dev) provide biochemical evidence for binding of variants containing ZNF domains with variants lacking it. And Baker et al (2015, PLOS Genet) provide genetic evidence (Fig. 5) that variants with different ZNF domains can bind to each other.

July 3, 2019

RE: Life Science Alliance Manuscript #LSA-2018-00291-TRRR

Dr. Irene Tiemann-Boege
Johannes Kepler University, Linz, Austria
Institute of Biophysics
Gruberstrasse 40
Linz 4020
Austria

Dear Dr. Tiemann-Boege,

Thank you for submitting your Research Article entitled "PRDM9 forms a trimer by interactions within the zinc finger array". It is a pleasure to let you know that your manuscript is now accepted for publication in Life Science Alliance. Congratulations on this interesting work.

DISTRIBUTION OF MATERIALS:

Again, congratulations on a very nice paper. I hope you found the review process to be constructive and are pleased with how the manuscript was handled editorially. We look forward to future exciting submissions from your lab.

Sincerely,
